# 🧑‍🔬 OSCAgent: Closing the Loop in Organic Solar Cell Discovery with LLM Agents

## Abstract

Organic solar cells (OSCs) hold great promise for sustainable energy, but discovering high-performance materials is time-consuming and costly. Existing molecular generation methods can aid the design of OSC molecules, but they are mostly confined to optimizing known backbones and lack effective use of domain-specific chemical knowledge, often leading to unrealistic candidates. In this paper, we introduce OSCAgent, a multi-agent framework for OSC molecular discovery that unifies retrieval-augmented design, molecular generation, and systematic evaluation into a continuously improving pipeline, without requiring additional human intervention. OSCAgent comprises three collaborative agents. The Planner retrieves knowledge from literature-curated molecules and prior candidates to guide design directions. The Generator proposes new OSC acceptors aligned with these plans. The Experimenter performs comprehensive evaluation of candidate molecules and provides feedback for refinement. Experiments show that OSCAgent produces chemically valid, synthetically accessible OSC molecules and achieves superior predicted performance compared to both traditional and large language model (LLM)-only baselines. Representative results demonstrate that some candidates achieve predicted efficiencies approaching 18%. The code will be publicly available.

## 1 Introduction

As the global demand for renewable energy continues to grow, organic solar cells (OSCs) have garnered significant attention for their capability to efficiently convert sunlight into electricity (Wang et al., 2016; Chen, 2019). Beyond their photovoltaic performance, OSCs provide unique advantages such as mechanical flexibility, lightweight processing, and low-cost fabrication, making them promising alternatives to conventional silicon-based solar technologies (Dennler et al., 2009).

Despite these advantages, the advancement of OSCs has been substantially constrained by the reliance on trial-and-error discovery strategies. The development of new materials often requires complex multi-step synthesis and costly characterization, making the exploration of chemical space both time-consuming and resource-intensive (Sun et al., 2019a). In this paradigm, the identification of high-performance OSC candidates remains inefficient, and systematic design principles are still underdeveloped. Consequently, it is both promising and imperative to explore data-driven approaches for advancing OSC molecular discovery.

Recently, artificial intelligence has been extensively applied across various scientific disciplines, including biology (Jumper et al., 2021; Corso et al., 2022), physics (Li et al., 2020; Wu et al., 2024), and chemistry (Baum et al., 2021; Zhou et al., 2023). Building on these advances, research on OSCs has also begun to embrace artificial intelligence as a key enabler of progress (Sun et al., 2019a). Most existing efforts (Sahu et al., 2018; Wu et al., 2020; Nagasawa et al., 2018; Meftahi et al., 2020; Nguyen et al., 2024; Ding et al., 2025) have concentrated on applying machine learning models to predict molecular properties, thereby assisting in the screening and analysis of candidate materials. Despite promising results, the discovery of new molecules still largely relies on the expertise and intuition of domain scientists. To alleviate this challenge, some researchers attempt to design molecules using fragment recombination, variational autoencoders (VAEs) (Sun et al., 2024), or genetic algorithms (Cao et al., 2025; Greenstein & Hutchison, 2023). However, these

methods typically focus on optimizing the backbones of known molecules and lack the ability to effectively integrate expert chemical knowledge, failing to generate valid or diverse OSC candidates. Furthermore, experimentally verified high-performance OSC molecules are scarce. This scarcity makes it difficult to train robust and generalizable generative models.

In this work, we propose OSCAgent, a multi-agent framework that closes the loop for discovery of OSC acceptor molecules, without requiring additional human intervention. By leveraging the knowledge of large language models (LLMs) in chemistry and materials science, OSCAgent can explore broader regions of chemical space and generate novel candidates beyond existing structures. The framework consists of three collaborative agents. The Planner employs a retrieval-augmented (Lewis et al., 2020) strategy, retrieving diverse, experimentally confirmed high-performance molecules together with dynamically updated top candidates to ground its reasoning and guide design directions. The Generator follows these directions to propose novel candidate molecules, while the Experimenter performs comprehensive evaluation using cheminformatics and machine learning tools. Candidate molecules are assessed not only for predicted power conversion efficiency (PCE) (Scharber et al., 2006), but also for synthetic accessibility (Ertl & Schuffenhauer, 2009) and electronic properties. Based on these results, the candidate database is dynamically updated to inform subsequent designs. By integrating these components, OSCAgent creates a continuously learning environment that enables the system to adapt and improve over time.

In summary, our main contributions are as follows:

- We introduce OSCAgent, an LLM-driven multi-agent system for OSC acceptor discovery, demonstrating its effectiveness in generating higher-quality candidate molecules.
- We enhance candidate evaluation with a multimodal PCE predictor incorporating uncertainty quantification, which significantly improves prediction accuracy and provides reliable feedback for guiding molecular design.
- We introduce a retrieval-augmented design strategy that combines experimentally validated, diverse molecules with dynamically updated top candidates, enabling informed exploration that balances novelty and feasibility.

## 2 RELATED WORK

**Artificial Intelligence for OSCs.** Artificial intelligence has recently emerged as a powerful tool for OSCs, particularly in predicting PCE and guiding molecular design. Early studies primarily relied on handcrafted molecular fingerprints, using traditional machine learning algorithms to predict the PCE of OSC materials (Mahmood & Wang, 2021; Zhao et al., 2020). Inspired by the success of graph neural networks (GNNs) in molecular property prediction, Eibeck et al. (Eibeck et al., 2021) explored their application to OSC property prediction. Building on this idea, GLaD (Nguyen et al., 2024) adopts a multimodal approach that integrates molecular graph representations with textual descriptors to further improve PCE prediction. More recently, RingFormer (Ding et al., 2025) enhances the ability of GNNs to capture ring-specific features, enabling more accurate modeling of the structural motifs that are critical in OSC molecules.

Beyond predictive modeling, artificial intelligence has also been applied to molecular design. Cao et al. (Cao et al., 2025) combined machine learning with genetic algorithms to enable efficient OSC molecular design, while DeepAcceptor (Sun et al., 2024) employed a VAE (Kingma et al., 2019) framework to generate novel OSC candidate molecules. Our work introduces an LLM-based multi-agent framework that enables closed-loop discovery of novel OSC acceptors.

**LLM Agents for Science.** Large language model (LLM) agents have recently found broad applications across scientific domains, including biology (Roohani et al., 2024; Su et al., 2025; Ghafarollahi & Buehler, 2024), chemistry (Tang et al., 2025; M. Bran et al., 2024; Ruan et al., 2024), materials science (Zhang et al., 2024; Hu et al., 2025a; Tian et al., 2025), and physics (Wuwu et al., 2025; Liu et al., 2024). PhenoGraph (Niyakan & Qian, 2025) leverages LLM agents to automate spatial transcriptomics data analysis. ChemAgent (Tang et al., 2025) enhances chemical reasoning by incorporating diverse types and functionalities of memory. OSDA Agent (Hu et al., 2025b) focuses on zeolite synthesis, integrating molecular generation, quantum evaluation, and feedback to identify suitable organic structure-directing agents. PINNsAgent (Wuwu et al., 2025) develops an automated framework for tuning physics-informed neural networks, significantly improving search efficiency.

Our work follows this emerging agentic paradigm but tailors it specifically to organic solar cell (OSC) acceptors molecular design. OSCAgent adapts the general principles of retrieval-guided planning, iterative feedback, and chemistry-aware generation to the challenges of discovering high-performance OSC acceptor molecules, thereby positioning our system as a domain-specialized instance of LLM-based scientific agents.

## 3 PRELIMINARIES AND DATA DESCRIPTION

### 3.1 ORGANIC SOLAR CELLS (OSCS)

OSCs are a class of photovoltaic devices that use organic molecules or polymers as active materials to convert sunlight into electricity. Their operation is based on a donor–acceptor structure: the donor absorbs light and generates electron–hole pairs, which are separated at the donor–acceptor interface, with electrons transported through the acceptor material and holes through the donor material. The molecular structures of OSC components determine key properties such as light absorption and charge transport, which directly influence the overall PCE (Scharber et al., 2006). Recent advances in non-fullerene acceptors have significantly improved OSC efficiency (Zhu et al., 2021), and in line with previous work (Sun et al., 2024), our study focuses on the design of OSC acceptor molecules.

### 3.2 POWER CONVERSION EFFICIENCY (PCE)

PCE (Scharber et al., 2006) is the key metric for assessing the performance of OSCs, measuring the proportion of incident solar energy converted into electrical power. PCE is defined as

$$\text{PCE} = \frac{V_{\text{OC}} \times J_{\text{SC}} \times \text{FF}}{P_{\text{in}}} \times 100\%,$$

where $V_{\text{OC}}$ is the open-circuit voltage, $J_{\text{SC}}$ is the short-circuit current density, FF is the fill factor, and $P_{\text{in}}$ is the incident light power density. In practice, PCE is the principal benchmark for material screening and device optimization, and has also become a central prediction and optimization target in recent machine learning studies of OSCs (Wu et al., 2020). Among the factors influencing PCE, the frontier orbital energies—the highest occupied molecular orbital (HOMO) and the lowest unoccupied molecular orbital (LUMO)—play a critical role. Their relative alignment governs exciton dissociation, charge transfer, and the achievable $V_{\text{OC}}$. Since HOMO and LUMO levels are strongly correlated with PCE, we incorporate them as auxiliary prediction tasks to support PCE training.

### 3.3 DATA

We utilized the OSC experimental PCE dataset curated by Sun et al. (Sun et al., 2024), which comprises 1,027 OSC acceptor molecules collected from 508 published articles. In cases where a molecule appeared in multiple studies, the maximum reported PCE value was retained to ensure consistency across entries. In addition, for each acceptor molecule, we calculated its synthetic accessibility score (SAscore) (Ertl & Schuffenhauer, 2009), which quantifies the estimated feasibility of synthesis on a scale from 1 (easy to synthesize) to 10 (difficult to synthesize).

Given the scarcity of experimentally available OSC molecules, we additionally employed the computational dataset constructed by Lopez et al. (Lopez et al., 2017) as a supplement. This dataset originates from the Harvard Clean Energy Project (CEP) (Hachmann et al., 2011) and contains 51,256 molecules that were generated and evaluated through high-throughput density functional theory (DFT) screening to establish a curated library of potential OSC materials.

## 4 METHODOLOGY

In this section, we introduce OSCAgent, our multi-agent framework for OSC discovery. Section 4.1 outlines the overall architecture, Section 4.2 describes the evaluation system that provides comprehensive feedback on candidate molecules, and Section 4.3 details the retrieval-augmented design strategy that integrates these evaluations to guide molecular generation.

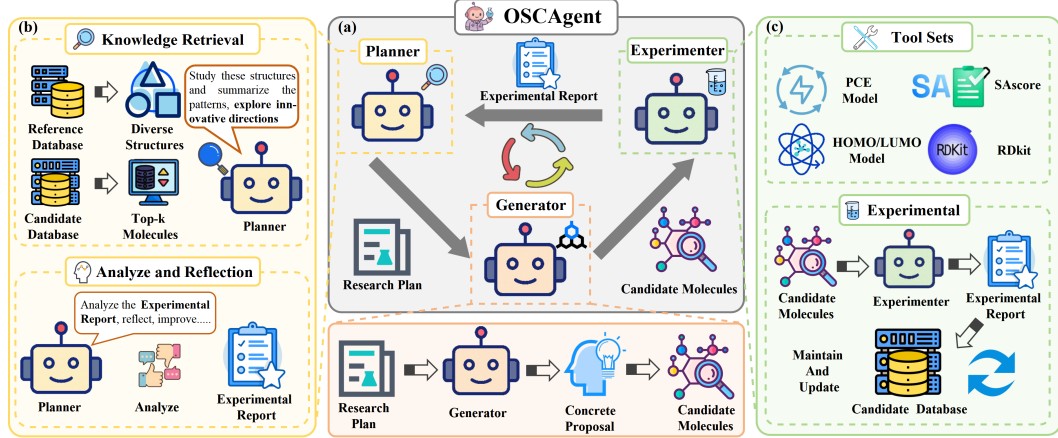

Figure 1: OSCAgent Framework. (a) The pipeline consists of three agents: Planner, Generator, and Experimenter. (b) The Planner retrieves knowledge from the Reference and Candidate databases to analyze diverse structures, summarize patterns, and propose research plans. It further reflects on evaluation reports to refine subsequent directions. (c) The Experimenter conducts a comprehensive evaluation using tools and maintains the Candidate database by incorporating promising molecules for future iterations.

## 4.1 OSCAGENT ARCHITECTURE

We propose OSCAgent, an LLM-based multi-agent framework for the discovery of acceptors for OSCs. The overall architecture is illustrated in Figure 1. It comprises three collaborative agents Planner, Generator, and Experimenter that coordinate strategic guidance, candidate proposal, and comprehensive evaluation.

The Planner serves as the strategic coordinator of OSCAgent, responsible for guiding molecular exploration through a retrieval-augmented approach. It retrieves molecules from two complementary sources: experimentally confirmed high-performance OSC molecules from the literature and a dynamically updated library of top-performing candidates (Section 4.3). Beyond retrieval, the Planner synthesizes chemical knowledge and patterns from prior exploration to summarize insights and formulate research plans that direct the Generator. Importantly, it also reflects on the evaluation results returned by the Experimenter, revising and refining its strategies to improve subsequent design cycles. In this way, the Planner ensures that candidate generation is both scientifically grounded and adaptively responsive to experimental feedback.

The Generator operates under the Planner's guidance, learning and understanding the research directions and translating them into concrete molecular designs and specific modification strategies. It produces candidate molecules that capture the targeted structural adjustments. Rather than engaging in unconstrained generation, the Generator emphasizes chemical plausibility and synthetic accessibility, ensuring that the proposed candidates are both scientifically meaningful and consistent with the Planner's objectives.

The Experimenter is responsible for the systematic evaluation of candidate molecules. We establish a comprehensive assessment pipeline that covers three complementary aspects: efficiency, assessed by a multimodal PCE predictor with uncertainty quantification; synthetic accessibility, which reflects the feasibility of molecular preparation; and orbital energy consistency, evaluated through machine learning models that predict HOMO/LUMO levels to ensure candidates remain within empirically observed ranges (details in Section 4.2). Based on these criteria, the Experimenter compiles structured reports that summarize candidate performance. In addition, promising molecules are added to the dynamically updated candidate library, enabling the Planner to build on high-performing structures in subsequent design iterations.

By integrating retrieval-augmented planning, guided molecular generation, and comprehensive evaluation, OSCAgent establishes a closed-loop workflow for systematic exploration of chemical space. Through multi-agent collaboration, the framework continuously adapts and improves, enabling ef-

ficient discovery of high-performance OSC acceptor molecules (see Appendix F.5, Figure 7 for a representative run of OSCAgent).

## 4.2 COMPREHENSIVE EVALUATION FRAMEWORK FOR OSC MOLECULAR DISCOVERY

We establish a comprehensive evaluation framework to systematically assess candidate molecules along three dimensions: efficiency, synthetic accessibility, and physical feasibility. Section 4.2.1 introduces the core of this framework, the PCE prediction tool, while Section 4.2.2 presents the other complementary tools.

### 4.2.1 UNCERTAINTY-AWARE MULTI-MODAL PCE PREDICTION

PCE is the most critical metric for evaluating the performance of OSCs, and thus accurate prediction is essential for guiding molecular discovery. Previous methods have primarily relied on training PCE prediction models directly on experimental datasets. However, this approach suffers from two major limitations. First, the scarcity of experimental data leads to poor model performance when trained from scratch. Second, because PCE values are experimentally measured, they are subject to variability across laboratories, experimental conditions, and fabrication processes, making PCE prediction inherently uncertain.

To fully exploit the available training resources, our model (see Figure 2) is first pretrained on the Lopez dataset (Lopez et al., 2017). During pretraining, we adopt a contrastive learning objective to align the graph and SMILES embeddings of the same molecule while separating those of different molecules. In addition, we integrate a multi-task objective by predicting the LUMO energy level, which provides an auxiliary signal strongly correlated with photovoltaic performance. This joint training strategy encourages the model to learn chemically meaningful representations that transfer effectively to downstream PCE prediction.

During fine-tuning on limited experimental data, we additionally incorporate handcrafted molecular descriptors (*e.g.*, Morgan, MACCS fingerprints), which are processed through a Mixture-of-Experts (MoE) (Shazeer et al., 2017) encoder and concatenated with the graph- and SMILES-based representations. Fingerprints offer complementary structural cues that help the model capture useful molecular patterns, especially in low-data regimes.

Given the inherent noise and variability of experimental measurements, we explicitly incorporate uncertainty quantification into our framework by treating PCE as a random variable rather than a fixed scalar (Kendall & Gal, 2017; Chen et al., 2024). Specifically, the predictor outputs both a mean $\mu(x)$ and a variance $\sigma(x)^2$ for each molecule, modeling PCE as a Gaussian distribution. The uncertainty-aware loss is defined as

$$L_{UQ} = \frac{1}{N} \sum_{i=1}^{N} \left( \frac{\|y_i - \mu(x_i)\|^2}{2\sigma(x_i)^2} + \frac{1}{2} \log \sigma(x_i)^2 \right),$$

which simultaneously minimizes prediction error and calibrates predictive variance. This probabilistic formulation enables the model to quantify its confidence, thereby reducing the risk of over-reliance on uncertain predictions. Model architecture and training details are in Appendix E.

In summary, our framework combines multimodal pretraining with uncertainty-aware modeling, yielding a more reliable predictor for molecular discovery in OSCs.

### 4.2.2 OTHER EVALUATION TOOLS

Beyond PCE prediction, OSCAgent incorporates complementary cheminformatics and property-evaluation tools to provide a holistic assessment of candidate molecules.

**SMILES Validation.** We employ RDKit (Landrum, 2013) to parse and sanitize SMILES strings, ensuring that all generated molecules are syntactically valid and chemically feasible.

**Synthetic Accessibility.** We compute the synthetic accessibility score (SAscore) (Ertl & Schuffenhauer, 2009) to estimate the ease of synthesis, using it as a constraint to ensure that proposed molecules remain practically synthesizable.

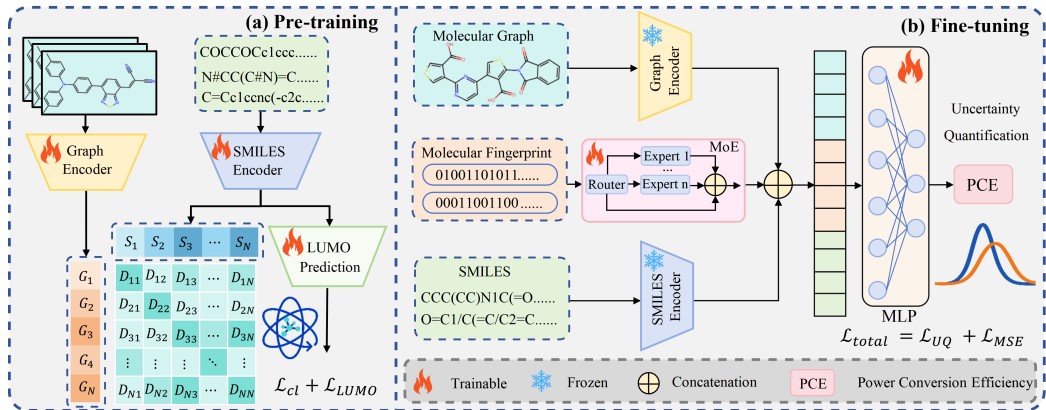

Figure 2: Details of the PCE prediction model. (a) Pre-training: Graph and SMILES encoders are jointly trained with contrastive learning and an auxiliary LUMO prediction task. To simplify the illustration, the LUMO prediction module of the graph branch is omitted from the figure. (b) Fine-tuning: Graph and SMILES embeddings are fused with molecular fingerprints via an MoE encoder, and the model is trained for PCE prediction with uncertainty quantification.

**Electronic Properties.** We train predictive models for HOMO and LUMO energy levels and use them as reference indicators, encouraging values to remain within empirically observed ranges of real OSC molecules while discouraging deviations that are too large. Further implementation details are provided in Appendix F.2.

Together, these tools complement PCE prediction by ensuring that candidate molecules are efficient, chemically valid, synthetically accessible, and aligned with key orbital energy characteristics. The Experimenter systematically records their outputs to guide subsequent decisions.

## 4.3 RETRIEVAL-AUGMENTED STRATEGY FOR OSC MOLECULES

To enhance the design capability of OSCAgent, we adopt a retrieval-augmented (Lewis et al., 2020) strategy tailored for OSC molecules. Instead of relying solely on the general knowledge captured during LLM pretraining, the Planner retrieves information from two complementary external sources.

A reference database was constructed from experimentally validated high-performance OSC molecules reported in the literature. For retrieval, molecules in this database are encoded into molecular fingerprints, and structural similarity is assessed using the Tanimoto distance (Tanimoto, 1958). To maximize coverage of the OSC chemical space, a $K$-center greedy algorithm (Gonzalez, 1985) is employed to select structurally diverse representatives. This ensures that the retrieved set consists of structurally diverse yet experimentally confirmed high-performance OSC molecules. Further implementation details are provided in Appendix D.1.

In parallel, a Candidate database is dynamically updated based on the evaluations provided by the Experimenter. From this database, the Top-$K$ molecules are retrieved using a carefully designed scoring function,

$$\text{Score}(m) = \text{PCE}(m) - \text{SAscore}(m) + f(\text{HOMO}(m), \text{LUMO}(m)),$$

where $m$ denotes a candidate molecule, PCE denotes predicted efficiency, SAscore penalizes synthetic difficulty, and $f(\text{HOMO}, \text{LUMO})$ is formulated as an interval-based reward function that encourages orbital energies to remain within empirically observed ranges of real OSC molecules, while discouraging large deviations. Further details are provided in Appendix D.2.

By integrating experimentally confirmed chemical knowledge from the literature with feedback derived from prior exploration, this retrieval-augmented design strategy enables OSCAgent to generate OSC acceptor molecules that are efficient, practically synthesizable, and structurally diverse.

## 5 EXPERIMENTS

In this section, we present the experimental evaluation of OSCAgent. Section 5.1 describes the experimental setup, including datasets, baselines, and implementation details. Section 5.2 reports the results of OSC molecular design, with both quantitative comparisons and ablation studies, as well as case studies. Section 5.3 presents the results of PCE prediction.

### 5.1 EXPERIMENTAL SETUP

In the OSCAgent framework, the Planner, Generator, and Experimenter all invoke the GPT-5 API. The predictive models used by the Experimenter for predicting PCE, HOMO/LUMO are trained on the Lopez dataset (Lopez et al., 2017) together with the experimental dataset collected by Sun et al. (Sun et al., 2024).

We compare OSCAgent against two categories of baselines: traditional molecular generation methods and LLM–based approaches. For traditional methods, we adopt the BRICS (Degen et al., 2008) and VAE (Kingma et al., 2019) strategies from DeepAcceptor (Sun et al., 2024). BRICS fragments existing molecules according to predefined chemical rules and recombines these fragments to generate new candidates, while VAE encodes molecules into a continuous latent space and decodes them into novel structures. For language model-based approaches, we evaluate BioT5 (Pei et al., 2023), a text-to-molecule generation model pretrained on large-scale molecular corpora, and Few-shot + Direct Reasoning, where a curated set of high-performance OSC molecules is provided as in-context examples, using the same LLM backbone as OSCAgent but without multi-agent orchestration.

### 5.2 RESULTS OF OSC MOLECULE DESIGN

We evaluate the effectiveness of OSCAgent by comparing it with traditional molecular generation methods (BRICS and VAE) and LLM-based methods (BioT5 and Few-shot + Direct Reasoning). For fairness, the GPT-5 models in the Few-shot setting were provided with the same high-performance OSC molecules that served as prompts in OSCAgent.

To assess performance, we use seven established metrics (Appendix A) covering diversity, effectiveness, and distributional alignment. Diversity is quantified by uniqueness—the proportion of distinct molecules among generated candidates—and novelty—the fraction of molecules absent from the existing dataset. Effectiveness is evaluated by validity, $i.e.$, the proportion of chemically valid molecules that satisfy design criteria (PCE $> 10\%$, SAscore $< 8$), and by average PCE, the mean predicted efficiency of generated molecules. Finally, distributional similarity to real high-performance OSCs is assessed by the Sinkhorn–Wasserstein distance (Cuturi, 2013) computed on four molecular fingerprints (Morgan, MACCS, RDK, and ECFP6), which is further expressed as a similarity score, with larger values indicating stronger alignment with the true molecular distribution.

Table 1: Performance comparison of different molecular generation methods. Best values are shown in **bold**, and second-best are underlined. Our method (OSCAgent) is highlighted at the bottom.

| Method | Diversity | | Molecular Quality | | Distribution Similarity | | | |
|---|---|---|---|---|---|---|---|---|
| | Uniqueness ↑ | Novelty ↑ | Validity ↑ | Avg. PCE (%)↑ | Morgan ↑ | MACCS ↑ | RDK ↑ | ECFP6 ↑ |
| BRICS | 0.871 | **1.000** | 0.049 | 6.461 | 0.337 | 0.694 | 0.681 | 0.276 |
| VAE | **0.919** | **1.000** | 0.002 | 4.967 | 0.219 | 0.611 | 0.663 | 0.167 |
| BioT5 | 0.802 | **1.000** | 0.000 | 3.573 | 0.132 | 0.424 | 0.462 | 0.114 |
| Few-shot | 0.709 | 0.989 | 0.283 | 9.249 | 0.394 | 0.693 | 0.684 | 0.326 |
| **OSCAgent** | 0.893 | **1.000** | **0.705** | **14.59** | **0.475** | **0.748** | **0.857** | **0.395** |

As shown in Table 1, traditional molecular generation approaches such as BRICS and VAE rely on fragment recombination or latent space sampling, essentially exploring variations of existing structures without clear design guidance. Consequently, most generated candidates are chemically infeasible or exhibit poor performance. Among language model-based methods, BioT5 faces significant limitations because it lacks OSC-specific chemical knowledge, making it difficult to generate candidates that meet performance requirements. In contrast, general-purpose LLMs like GPT, guided by

few-shot prompting with strong examples, can produce more reasonable molecules. Nevertheless, without the integration of specialized chemical tools, the few-shot approach still suffers from clear shortcomings in both accuracy and diversity.

In contrast, our proposed OSCAgent demonstrates consistent superiority over all baseline methods. By combining the chemical knowledge encoded in LLMs with domain-specific evaluation tools and knowledge-augmented design strategies, OSCAgent is able to generate OSC molecules that are both chemically valid and performance-oriented. As shown in Table 1, it achieves the best validity and the highest average PCE, while also obtaining the strongest distributional similarity to real high-performance molecules across multiple fingerprinting metrics. These results highlight that OSCAgent not only produces feasible and diverse structures, but also identifies candidates with superior photovoltaic potential, underscoring the advantage of a closed-loop, knowledge-augmented multi-agent framework for OSC discovery. In addition, we also evaluate OSCAgent on open-source LLMs, further confirming its effectiveness (see Appendix F.1).

### 5.2.1 Ablation Study

To better understand the contribution of each component in the OSCAgent framework, we conduct ablation experiments along two dimensions.

**(1) Without Retrieval-Augmented Strategy.** In this setting, the Planner relies solely on fixed few-shot prompts and the general knowledge acquired during LLM pretraining, without retrieving molecules from either the Reference Database of experimentally confirmed OSCs or the Candidate Database updated from prior feedback. As shown in Table 2, this results in a clear decline in performance, with both predicted PCE and distributional similarity scores dropping compared to the full model. These findings suggest that static prompting alone is insufficient, and that retrieval-augmented knowledge from literature-derived references and prior experience is crucial for guiding the design process toward chemically realistic and structurally diverse candidates.

**(2) Without Experimenter.** In this setting, the Planner receives no feedback from chemical evaluation tools and must rely entirely on its own pretrained knowledge and heuristic judgment when formulating design plans. Without this feedback loop, the design process loses critical guidance: although structural diversity remains comparable, candidate validity and predicted efficiency degrade noticeably. These results underscore the pivotal role of the Experimenter in providing systematic chemical assessment and feedback, which steers exploration toward chemically feasible and high-performing regions of the design space.

Overall, these results demonstrate that both the Retrieval-Augmented Strategy and the Experimenter are indispensable. The former enriches the Planner's reasoning with literature knowledge and prior feedback, while the latter provides systematic chemical evaluation to refine candidate quality. Together, they enable OSCAgent to achieve more effective and reliable molecular discovery.

Table 2: Ablation study of OSCAgent. We report diversity (uniqueness), molecular quality (validity and average PCE), and distributional similarity scores computed with four fingerprints (Morgan, MACCS, RDK, ECFP6). Removing either the Retrieval-Augmented Strategy or the Experimenter results in a clear degradation of performance across multiple metrics.

| Method | Diversity | Molecular Quality | | Distribution Similarity | | | |
|---|---|---|---|---|---|---|---|
| | Uniqueness ↑ | Validity ↑ | Avg. PCE(%) ↑ | Morgan ↑ | MACCS ↑ | RDK ↑ | ECFP6 ↑ |
| OSCAgent (full) | **0.893** | **0.705** | **14.59** | **0.475** | **0.748** | **0.858** | **0.395** |
| w/o Retrieval-Aug. Strategy | 0.813 | 0.518 | 10.41 | 0.414 | 0.723 | 0.817 | 0.356 |
| w/o Experimenter | 0.847 | 0.387 | 13.21 | 0.457 | 0.735 | 0.845 | 0.370 |

### 5.2.2 Case Studies of Generated OSC Molecules

To qualitatively evaluate the design capability of OSCAgent, we showcase three representative OSC acceptor molecules generated by the framework in Figure 3. These candidates preserve hallmark structural motifs of high-performance OSCs (such as extended $\pi$-conjugation, and electron-

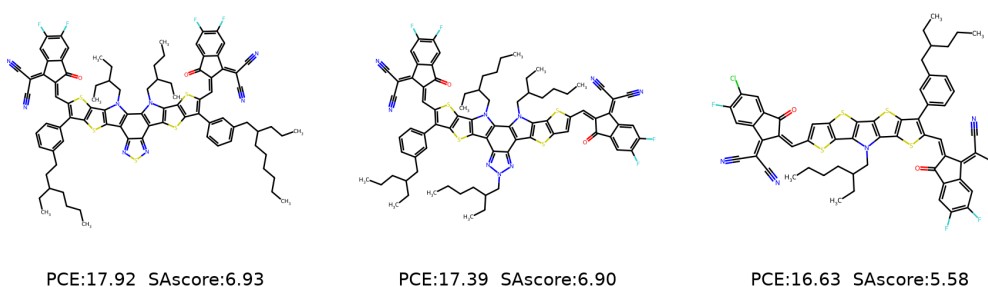

PCE:17.92 SAscore:6.93        PCE:17.39 SAscore:6.90        PCE:16.63 SAscore:5.58

Figure 3: Representative OSC molecules generated by OSCAgent

withdrawing cyano groups) while simultaneously introducing novel structural variations. The predicted results indicate that their PCE exceed 16%, with synthetic accessibility scores (SAscore) consistently below 8.0, suggesting a balance between efficiency and feasibility. These case studies highlight the ability of OSCAgent to generate OSC molecules that effectively trade off novelty, synthetic practicality, and photovoltaic performance. Additional examples of designed molecules, together with demonstrations of multi-agent collaboration during the design process, are provided in Appendix F.5.

### 5.3 RESULTS OF PCE PREDICTION

To assess the effectiveness of our PCE predictor, we evaluate it on the OSC experimental dataset (Sun et al., 2024), comparing with baseline models and ablation studies on key components. Table 3 reports the performance in terms of $R^2$ and MAE.

For baselines, we cover both traditional machine learning methods and recent neural architectures. As a classical approach, we adopt Morgan molecular fingerprints combined with a Random Forest (RF) regressor, which has long been used for PCE prediction. Among neural models, we include a Transformer applied to molecular SMILES sequences, an MPNN operating on molecular graphs, and two recent graph transformer variants, RingFormer (Ding et al., 2025) and GRIT (Ma et al., 2023), which are specifically designed to capture structural features of molecular graphs. In addition, we evaluate abcBERT (Sun et al., 2024), a model that performs pretraining exclusively on the molecular graph modality.

Table 3: Performance comparison and ablation study for PCE prediction.

| Method | $R^2 \uparrow$ | MAE $\downarrow$ |
| --- | --- | --- |
| Morgan + RF | 0.649 | 1.875 |
| Transformer | 0.554 | 2.128 |
| MPNN | 0.589 | 2.027 |
| Ringformer | 0.631 | 1.948 |
| GRIT | 0.643 | 1.901 |
| abcBERT | 0.668 | 1.781 |
| **Ours (full)** | **0.713** | **1.686** |
| w/o UQ | 0.681 | 1.776 |
| w/o Pretraining | 0.654 | 1.879 |
| w/o Handcrafted Feat. | 0.634 | 1.924 |

We observe that many recent neural architectures, despite their success in other molecular tasks, underperform on PCE prediction compared to the traditional fingerprint-based approach. This highlights the necessity of incorporating molecular fingerprints as complementary features in our design. Among baselines, abcBERT benefits from graph-level pretraining and shows competitive performance, yet it still lags behind our predictor. This shows the advantage of combining multimodal representations with uncertainty-aware learning in capturing the factors that determine PCE..

The ablation study confirms that uncertainty quantification, pretraining, and handcrafted molecular descriptors each make meaningful and complementary contributions to our framework. The integration of these components yields a predictor with markedly improved accuracy, providing OSCAgent with a more reliable foundation for downstream molecular design.

## 6 CONCLUSION

In this work, we introduced OSCAgent, a multi-agent framework for OSC discovery. OSCAgent generated chemically valid, synthetically accessible, and high-performing OSC acceptor candidates

that outperformed both traditional and LLM-based baselines. Looking ahead, we plan to extend the framework to broader classes of functional materials and integrate wet-lab feedback for tighter design–validation coupling.

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

## A EVALUATION METRICS

To comprehensively assess the quality of generated OSC molecules, we employ seven widely used metrics. Their detailed definitions are provided below.

**Uniqueness.** Uniqueness evaluates the diversity of generated molecules by measuring the fraction of distinct structures within the generated set. Formally,

$$\text{Uniqueness} = \frac{N_{\text{unique}}}{N_{\text{generated}}},$$

where $N_{\text{unique}}$ is the number of unique molecules and $N_{\text{generated}}$ is the total number of generated molecules. A higher Uniqueness score indicates that the model is able to generate a broader range of structurally diverse candidates rather than repeatedly producing the same molecules.

**Novelty.** Novelty measures the proportion of generated molecules that are not present in the reference dataset, thereby reflecting the ability to explore new chemical space. Formally,

$$\text{Novelty} = \frac{N_{\text{novel}}}{N_{\text{generated}}},$$

where $N_{\text{novel}}$ denotes the number of generated molecules absent from the reference dataset, and $N_{\text{generated}}$ is the total number of generated molecules. A higher Novelty score indicates a stronger capability to propose unseen OSC molecules rather than replicating known structures.

**Validity.** Validity quantifies the fraction of generated molecules that are not only chemically correct but also satisfy the design-oriented constraints. A molecule is considered valid if it (i) can be successfully parsed and sanitized by RDKit, ensuring syntactic and chemical correctness of the SMILES string, and (ii) meets the thresholds for photovoltaic efficiency and synthetic feasibility, specifically PCE > 10 and SAscore < 8. Formally,

$$\text{Validity} = \frac{N_{\text{valid}}}{N_{\text{generated}}},$$

where $N_{\text{valid}}$ denotes the number of molecules that are both chemically valid and meet the PCE/SAscore constraints, and $N_{\text{generated}}$ is the total number of generated molecules. A higher Validity score reflects the ability of the model to produce candidates that are simultaneously chemically sound and practically promising.

**Average PCE.** Average PCE measures the expected photovoltaic performance of the generated set by computing the mean predicted power conversion efficiency (PCE) across all valid molecules. Formally,

$$\text{Avg. PCE} = \frac{1}{N} \sum_{i=1}^{N} y_i,$$

where $y_i$ denotes the predicted PCE of molecule $i$, and $N$ is the total number of valid generated molecules. A higher Average PCE indicates that the generation strategy tends to produce candidates with stronger photovoltaic potential on average, reflecting its effectiveness in prioritizing high-performance OSC molecules.

**Distribution Similarity.** To evaluate how well the generated OSC molecules match real high-performance molecules (following the definition in Sun et al. (Sun et al., 2024), where high performance is characterized by PCE > 10% and SAscore < 8), we measure the similarity between their distributions. Each molecule is first encoded into a structural fingerprint (e.g., Morgan, MACCS, ECFP6, or RDKit). For two molecules $x_i$ and $y_j$, we define their dissimilarity using the **Tanimoto distance**,

$$C_{ij} = 1 - \text{Tanimoto}(x_i, y_j).$$

Given the generated distribution $P$ and the reference distribution $Q$, the **Wasserstein distance** $W(P, Q)$ is computed as the minimal average cost of transporting probability mass between them under the distance matrix $C = \{C_{ij}\}$.

We report a **Wasserstein similarity score**,

$$\text{Sim}(P, Q) = 1 - W(P, Q),$$

bounded between 0 and 1, where larger values indicate stronger alignment between generated and reference molecular distributions. Compared to simple statistics such as mean Tanimoto similarity, Wasserstein similarity provides a more comprehensive measure because it captures the *global structure* of distributions rather than just local pairwise overlaps.

**Tanimoto Similarity.** The **Tanimoto similarity** is the most widely used metric for comparing molecular fingerprints. For two molecules $x$ and $y$ with binary fingerprints $f_x, f_y \in \{0, 1\}^d$, it is defined as

$$\text{Tanimoto}(x, y) = \frac{|f_x \cap f_y|}{|f_x \cup f_y|} = \frac{c}{a + b - c},$$

where $a = \sum_i f_x(i)$, $b = \sum_i f_y(i)$, and $c = \sum_i f_x(i) f_y(i)$. This value ranges from 0 to 1, with 1 indicating identical fingerprints and 0 no shared features.

**Wasserstein Distance.** The **Wasserstein distance** (also known as the Earth Mover's Distance) (Cuturi, 2013) is a fundamental metric for comparing probability distributions. Given two distributions $P$ and $Q$ defined on a metric space with distance function $d(\cdot, \cdot)$, the 1-Wasserstein distance is defined as

$$W(P, Q) = \inf_{\pi \in \Pi(P, Q)} \sum_{i,j} \pi_{ij} \, d(x_i, y_j),$$

where $\Pi(P, Q)$ denotes the set of all couplings with marginals $P$ and $Q$. Intuitively, it measures the minimum "work" required to transform one distribution into the other, where $\pi_{ij}$ specifies the amount of probability mass moved from $x_i$ to $y_j$ and the cost is proportional to their ground distance $d(x_i, y_j)$.

# B   WORKING PRINCIPLE OF ORGANIC SOLAR CELL (OSC) MOLECULES

The working principle of organic solar cells (OSCs) can be described as a sequence of steps that convert sunlight into electricity. When the active materials absorb photons, they generate electron–hole pairs known as excitons. These excitons migrate to the interface between donor and acceptor molecules, where the energy offset between the two materials enables their separation into free charges: electrons move into the acceptor, while holes remain in the donor.

Once separated, the charges travel through their respective molecular pathways and are collected at the electrodes, resulting in an electric current. The overall efficiency of this process is determined by three key factors: the voltage, which depends on the energy alignment of donor and acceptor; the current, which reflects the material's ability to absorb light and generate free charges; and the fill factor, which measures how efficiently charges are transported without recombining. Together, these mechanisms illustrate how OSC molecules transform solar energy into usable electrical power.

# C   WHY OUR FRAMEWORK FOCUSES ON ACCEPTORS

Our work focuses on the design of high-potential acceptor molecules. Our objective is to identify acceptors with strong intrinsic photovoltaic potential, and our data formulation follows the same strategy used in prior studies such as DeepAcceptor (Sun et al., 2024), RingFormer (Ding et al., 2025), and earlier work by Sun et al. (Sun et al., 2019b). These studies model donors or acceptors independently rather than as donor–acceptor (D/A) pairs. They adopt the widely used "maximum PCE per molecule" labeling strategy, which enables the model to learn the inherent photovoltaic capability of an individual material. Our approach adheres to standard OSC data-curation practices and does not introduce any assumptions beyond those broadly accepted in the existing literature.

A practical consideration is that if we restrict the task to designing acceptors for a specific fixed donor, the amount of usable training data becomes drastically smaller, making it much more difficult for the model to learn meaningful patterns. By focusing on acceptor design alone, we can leverage all available data and learn more generalizable structure–property relationships (Sun et al., 2024).

# D  IMPLEMENTATION DETAILS OF RETRIEVAL-AUGMENTED STRATEGY

## D.1  REFERENCE DATABASE CONSTRUCTION AND RETRIEVAL

To support downstream decision-making, we construct a Reference Database that provides the agent with high-performing OSC acceptor molecules. These molecules are selected from the experimental dataset (Sun et al., 2024), ensuring that only high-performance candidates are included. The molecules are then standardized using RDKit, which involves canonical SMILES normalization, tautomer unification, and salt removal. For each molecule, Morgan fingerprints (radius = 2, 2048 bits) are generated, and pairwise similarity is quantified using the Tanimoto coefficient, with distance defined as

$$d(m_i, m_j) = 1 - \text{Tanimoto}(m_i, m_j).$$

**Diversity through $K$-center Greedy.**  To retrieve structurally diverse molecules and ensure broad coverage of chemical space, we adopt a $K$-center greedy selection strategy. The procedure begins by randomly selecting one molecule as the initial seed. The remaining $(K-1)$ representatives are then chosen iteratively such that each selected molecule maximizes its minimum distance to the already selected set $\mathcal{S}$:

$$m^* = \arg \max_{m \in \mathcal{M} \setminus \mathcal{S}} \min_{s \in \mathcal{S}} d(m, s).$$

In this way, the retrieved set captures not only high-performance molecules but also structurally dissimilar representatives, thereby promoting diversity and improving coverage of the underlying chemical space.

**Retrieval.**  To supply the Planner with informative references, we implement a retrieval module that balances chemical validity with structural diversity. As detailed in Algorithm 1, molecules in the Reference Database are first encoded into Morgan fingerprints, and pairwise similarities are computed using the Tanimoto coefficient. A $K$-center greedy selection strategy is then applied to retrieve a diverse set of molecules. The process starts by randomly selecting an initial molecule. Subsequently, the remaining $K-1$ molecules are selected iteratively by maximizing their dissimilarity to the already chosen set, ensuring that the final set of $K$ molecules is maximally diverse both within the set and in relation to the initial molecule. In practice, we set $K = 5$.

In this way, the Reference Database functions as a curated and diversity-enhanced knowledge base, equipping OSCAgent with robust references for molecular generation and evaluation.

## D.2  CANDIDATE DATABASE RETRIEVAL AND MAINTENANCE

The Candidate Database is maintained dynamically as the system explores new molecular candidates. During the execution of OSCAgent, the Experimenter continually evaluates newly proposed molecules and updates the Candidate Database with their predicted properties. After each iteration, all candidates are ranked according to a composite scoring function that balances efficiency, synthetic accessibility, and physical feasibility:

$$\text{Score}(m) = \text{PCE}(m) - \text{SAscore}(m) + f(\text{HOMO}(m), \text{LUMO}(m)).$$

Here, $\text{PCE}(m)$ denotes the predicted power conversion efficiency, obtained from surrogate models described in Section 4.2; $\text{SAscore}(m)$ is the synthetic accessibility score computed with RDKit; and $f(\text{HOMO}, \text{LUMO})$ is a feasibility adjustment based on frontier orbital energies. This adjustment reflects empirical evidence that effective OSC molecules exhibit orbital levels within specific ranges, ensuring favorable exciton dissociation and energy-level alignment (Scharber et al., 2006; Sun et al., 2024) (see Fig. 4).

Formally, the feasibility term is defined as

$$f(\text{HOMO}, \text{LUMO}) = \begin{cases} +\gamma, & \text{if } \text{HOMO}_{\min} \leq \text{HOMO} \leq \text{HOMO}_{\max}, \\ & \quad \text{and } \text{LUMO}_{\min} \leq \text{LUMO} \leq \text{LUMO}_{\max}, \\ -\delta, & \text{otherwise}, \end{cases}$$

---

**Algorithm 1:** K-Center Greedy Retrieval from Reference Database

---

**Input:** Reference Database $\mathcal{R}$ ; retrieval size $K$
**Output:** A set $S$ of $K$ diverse molecules; formatted examples (SMILES, PCE, SA, HOMO/LUMO)
**Step 1: Morgan Fingerprints**
    **foreach** *SMILES $m_i$ in $\mathcal{R}$* **do**
        Compute Morgan bit vector $f_i \leftarrow \text{Morgan}(m_i;\ \text{radius} = 2,\ \text{nBits} = 2048)$;
        If $x_i$ invalid, set $f_i \leftarrow \varnothing$ and mark record for exclusion.
    let $N \leftarrow |\mathcal{R}|$.
**Step 2: Similarity Matrix**
    Allocate $D^{(\text{sim})} \in \mathbb{R}^{N \times N}$; set diagonals $D_{ii}^{(\text{sim})} \leftarrow 1.0$.
    **for** $i \leftarrow 1$ **to** $N$ **do**
        Compute $\{D_{i,j}^{(\text{sim})}\}_{j=i+1}^{N} \leftarrow \text{BulkTanimotoSimilarity}(f_i, \{f_{i+1}, \ldots, f_N\})$;
        Set $D_{j,i}^{(\text{sim})} \leftarrow D_{i,j}^{(\text{sim})}$ for all $j > i$.
    Define distance matrix $D^{(\text{dist})} \leftarrow \mathbf{1} - D^{(\text{sim})}$.
**Step 3: K-Center Greedy**
    Pick the first center $i_0$ uniformly at random from $\{1, \ldots, N\}$; set $S \leftarrow \{i_0\}$.
    Initialize the minimum distance vector $\Delta \leftarrow D_{i_0,:}^{(\text{dist})}$.
    **while** $|S| < K$ **do**
        $i^\star \leftarrow \arg\max_{j \notin S} \Delta_j$
        $S \leftarrow S \cup \{i^\star\}$
        Update $\Delta \leftarrow \min\left(\Delta,\ D_{i^\star,:}^{(\text{dist})}\right)$.       // element-wise minimum;
**Step 4: Formatted Output**
    For each $i \in S$, output a line with:
        $\texttt{SMILES} = x_i,\quad \texttt{PCE} = \mathcal{R}[i, \texttt{pce}],\quad \texttt{SA} = \mathcal{R}[i, \texttt{sa\_score}],\quad \texttt{HOMO/LUMO}$
    $= (\mathcal{R}[i, \texttt{homo}],\ \mathcal{R}[i, \texttt{lumo}])$.
    **return** $S$ and the formatted examples.
*Note:* In Step 3, the update $\Delta \leftarrow \min(\Delta,\ D_{i^\star,:}^{(\text{dist})})$ is element-wise. For example, if $\Delta = [2.0, 5.0, 3.0, 4.0]$ and $D_{i^\star,:}^{(\text{dist})} = [1.0, 3.0, 2.0, 6.0]$, the updated $\Delta$ becomes $[1.0, 3.0, 2.0, 4.0]$.

---

with $\gamma > 0$ as a reward for molecules within feasible ranges and $\delta > 0$ as a penalty otherwise. Typical empirically derived orbital windows are:

$$\text{HOMO} \in [-6.0, -5.0] \text{ eV}, \quad \text{LUMO} \in [-4.5, -3.0] \text{ eV}.$$

In the experiments reported in this paper, we set $\gamma = \delta = 3$. This scoring rule ensures that candidates outside of realistic orbital windows are deprioritized, while those aligning with known OSC design principles are promoted.

At each retrieval step, the Top-$K$ molecules with the highest scores are selected from the Candidate Database. Since the Database is continually updated by the Experimenter, these exemplars evolve dynamically throughout the OSCAgent workflow. They serve as anchors in the Planner's prompts, enabling the agent to ground its reasoning in its own most promising discoveries while progressively refining molecular design. In practice, we set $K = 3$.

### D.3   RETRIEVAL-AUGMENTED PROMPTING

At the beginning of each design cycle, the Planner retrieves a set of reference molecules from the static Reference Database and a set of top-ranked candidates from the dynamic Candidate Database. Both sets are inserted into the Planner's prompt as contextual knowledge, ensuring that new molecular generation is informed by (i) experimentally validated high-performance motifs, and (ii) adaptive feedback from the system's prior explorations.

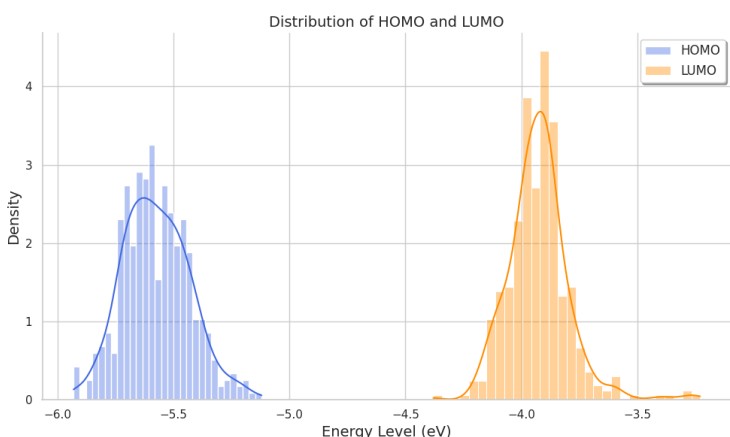

Figure 4: HOMO and LUMO energy level distributions of high-performance OSC acceptor molecules.

Table 4: Comparison of generation quality under different retrieval sizes $k$.

| Model | Morgan | MACCS | RDK | ECFP6 |
|---|---|---|---|---|
| $K = (5, 3)$ | 0.4752 | 0.7475 | 0.8576 | 0.3953 |
| $K = (7, 7)$ | 0.4572 | 0.7481 | 0.8587 | 0.3792 |
| $K = (2, 2)$ | 0.4216 | 0.7191 | 0.8209 | 0.3314 |

## D.4   K VALUE SELECTION

We evaluated different retrieval sizes in the initial RAG stage, as shown in Table 4. For small retrieval sizes (e.g., $K = 2$), the retrieved molecules lacked sufficient structural diversity and resulted in noticeably poorer distributional alignment between generated molecules and real OSC acceptors. For larger retrieval sizes (e.g., $K = 7$), performance remained stable, but increasing $K$ did not yield further improvements. Instead, the enlarged prompts—due to the inherently large and complex structures of OSC molecules—substantially increased token usage without providing additional benefit.

Based on these observations, we adopt intermediate values of $K$ to balance diversity and computational efficiency. Specifically, we use $K = 5$ for the Reference Database to retrieve a broad set of experimentally validated high-performance OSC molecules, and $K = 3$ for the Candidate Database to retrieve molecules that performed well in earlier iterations. Because experimentally reported molecules carry greater importance in guiding the design process, a larger $K$ is assigned to the Reference Database.

## E   MODEL DETAILS

To predict the power conversion efficiency (PCE) of OSC molecules, we employ a model that integrates multimodal molecular representations with an uncertainty-aware regression head. The training procedure is organized in two stages: (i) pretraining with contrastive and auxiliary objectives, and (ii) fine-tuning on experimental PCE measurements.

**Pretraining with Contrastive Learning.**   To integrate the long-range dependencies captured by SMILES sequences with the spatial and structural information provided by molecular graphs, we employ a multi-modal contrastive learning strategy. The central idea is that representations derived from different modalities should be close if they correspond to the same molecule, and far apart otherwise. Specifically, for a molecule $m_j$, we denote its SMILES-based representation as $f_S^j$ and its graph-based representation as $f_G^j$. The pair $(f_S^j, f_G^j)$ is treated as a positive pair, since they encode

the same molecule from two modalities. In contrast, $(f_S^j, f_G^k)$ or $(f_S^k, f_G^j)$ with $k \neq j$ are considered negative pairs, as they represent different molecules.

Formally, we adopt a symmetric InfoNCE loss (Oord et al., 2018) with a learnable temperature parameter $\tau \in \mathbb{R}_+$:

$$\mathcal{L}_{\text{cl}} = -\frac{1}{2} \log \frac{\exp\left(\langle f_G^j, f_S^j \rangle / \tau\right)}{\sum_{k=1}^{N} \exp\left(\langle f_G^j, f_S^k \rangle / \tau\right)} - \frac{1}{2} \log \frac{\exp\left(\langle f_G^j, f_S^j \rangle / \tau\right)}{\sum_{k=1}^{N} \exp\left(\langle f_G^k, f_S^j \rangle / \tau\right)}.$$

Table 5: Hyperparameter settings for pretraining and fine-tuning.

| Stage | Hyperparameter | Value |
|---|---|---|
| Pretraining | Graph hidden size | 64 |
| | Graph steps (MP / S2S) | 3 / 3 |
| | Graph pooling layers | 1 |
| | Graph output dim | 1024 |
| | SMILES embed dim | 512 |
| | SMILES layers | 4 |
| | SMILES context length | 512 |
| | SMILES output dim | 1024 |
| | Learning rate | $1 \times 10^{-4}$ |
| | Batch size | 128 |
| | $\lambda$ | 1.0 |
| Fine-tuning | MLP hidden size | 768 |
| | MoE layers / experts | 2 / 4 |
| | Dropout ratio | 0.3 |
| | Output dim | 512 |
| | Learning rate | $3 \times 10^{-4}$ |
| | Batch size | 128 |
| | Weight decay | $5 \times 10^{-5}$ |
| | $\alpha$ | 0.2 |

Here, $\langle \cdot, \cdot \rangle$ denotes the similarity function (implemented as the inner product of $\ell_2$-normalized vectors). The exponential transformation projects multi-modal embeddings into a shared space where InfoNCE encourages alignment of positive pairs and separation of negative pairs. During pretraining, both the SMILES encoder and the graph encoder are optimized jointly with this loss, and the resulting multimodal encoders are subsequently fine-tuned for power conversion efficiency (PCE) prediction.

**Auxiliary LUMO Prediction.** Frontier orbital energies, particularly the LUMO level, are closely tied to the electronic properties and performance of OSC molecules. To embed such physicochemical priors into representation learning, we design an auxiliary regression task that predicts LUMO energies from both graph-based and SMILES-based embeddings. Specifically, given molecular representations $\mathbf{z}^G$ and $\mathbf{z}^S$ produced by the graph and SMILES encoders, two regression heads output the predicted LUMO values $\hat{l}^G$ and $\hat{l}^S$. The auxiliary regression loss is defined as

$$\mathcal{L}_{\text{LUMO}} = \frac{1}{N} \sum_{i=1}^{N} \left( \|\hat{l}_i^G - l_i\|^2 + \|\hat{l}_i^S - l_i\|^2 \right),$$

where $l_i$ denotes the ground-truth LUMO energy. This auxiliary objective provides physically grounded supervision, guiding the shared representation space toward features that better capture the quantum-chemical characteristics of OSC molecules.

**Joint Pretraining Objective.** The overall pretraining loss combines the contrastive objective with the auxiliary regression term:

$$\mathcal{L}_{\text{pretrain}} = \mathcal{L}_{\text{cl}} + \lambda \cdot \mathcal{L}_{\text{LUMO}},$$

where the trade-off parameter $\lambda$ controls the relative strength of representation alignment and orbital-level supervision. By integrating structural alignment with physically meaningful prediction, the model acquires transferable representations that are more predictive of downstream PCE performance.

**Fine-tuning with Uncertainty Quantification.**    After pretraining, the model is fine-tuned on experimental OSC datasets (Sun et al., 2024). To enhance molecular representation during this stage, we concatenate the learned embeddings with molecular fingerprint features, which provide complementary cheminformatics descriptors. Instead of treating PCE as a deterministic scalar, we model it as a Gaussian random variable with mean $\mu(x)$ and variance $\sigma(x)^2$ predicted by the model. Training loss is:

$$\mathcal{L}_{\text{PCE}} = (1 - \alpha)\,\mathcal{L}_{\text{MSE}} + \alpha\,\mathcal{L}_{\text{UQ}}$$

$$= (1 - \alpha) \cdot \frac{1}{N} \sum_{i=1}^{N} \|y_i - \mu(x_i)\|^2 + \alpha \cdot \frac{1}{N} \sum_{i=1}^{N} \Big( \frac{\|y_i - \mu(x_i)\|^2}{2\sigma(x_i)^2} + \tfrac{1}{2} \log \sigma(x_i)^2 \Big).$$

which encourages accurate prediction while calibrating predictive uncertainty. This formulation allows downstream components of OSCAgent to weigh predictions according to confidence, reducing the risk of over-reliance on noisy estimates.

**Hyperparameters.**    We summarize the main hyperparameter settings for both pretraining and fine-tuning in Table 5.

# F    SUPPLEMENTARY EXPERIMENTS

To further validate the robustness and generality of our approach, we conduct additional experiments beyond the main results presented.

## F.1    EXPERIMENTS ON OPEN-SOURCE LLMS

In the main paper, our results are primarily based on a closed-source large language model, GPT-5. To examine whether the proposed framework generalizes to open-source backbones, we additionally conducted experiments with an open-source model, gpt-oss (Agarwal et al., 2025).

Table 6: Performance comparison between the proposed OSCAgent (gpt-oss) and a few-shot baseline using the open-source gpt-oss model. Reported metrics include molecular similarity scores (Morgan, MACCS, RDK, ECFP6), validity (Val), and average predicted PCE.

| Method | Morgan ↑ | MACCS ↑ | RDK ↑ | ECFP6 ↑ | Validity ↑ | Avg. PCE(%) ↑ |
|---|---|---|---|---|---|---|
| OSCAgent (gpt-oss) | **0.426** | **0.748** | **0.798** | **0.334** | **0.426** | **12.44** |
| Few-shot (gpt-oss) | 0.340 | 0.617 | 0.650 | 0.290 | 0.235 | 6.18 |

As shown in Table 6, our agentic framework (OSCAgent) significantly outperforms the few-shot baseline when applied to the open-source gpt-oss model. In particular, OSCAgent achieves higher molecular similarity scores across all fingerprint metrics, as well as improvements in both validity and average predicted PCE. These results indicate that our approach can substantially enhance the quality of OSC molecule design even under an open-source backbone. While the overall performance of gpt-oss remains below that of the closed-source GPT-5 reported in the main text, the consistent gains highlight the potential of our method to generalize across different language model families and to benefit from future advances in open-source LLMs.

## F.2    MULTI-MODAL HOMO/LUMO PREDICTION MODEL

To accurately estimate the HOMO and LUMO energy levels of candidate molecules, we construct a multimodal prediction model that combines molecular graph representations with SMILES-based embeddings. The model is first pretrained on the large-scale Lopez dataset and subsequently fine-tuned on experimental datasets with measured HOMO and LUMO values.

The pretraining objective is similar to the joint pretraining objective described in Appendix E, defined as

$$\mathcal{L}_{\text{pretrain}} = \mathcal{L}_{\text{cl}} + \lambda \cdot \mathcal{L}_{\text{HOMO/LUMO}},$$

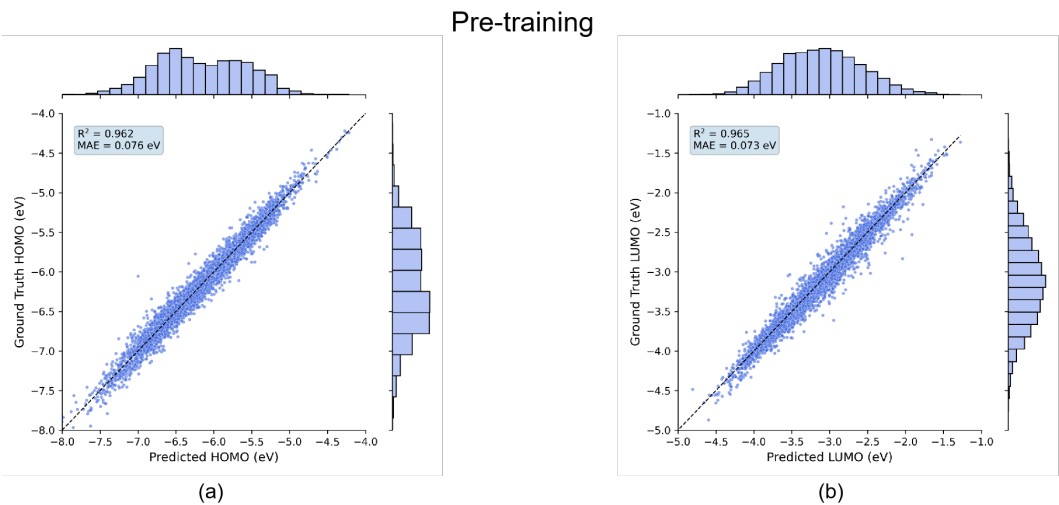

Figure 5: Pretraining performance of HOMO/LUMO prediction.

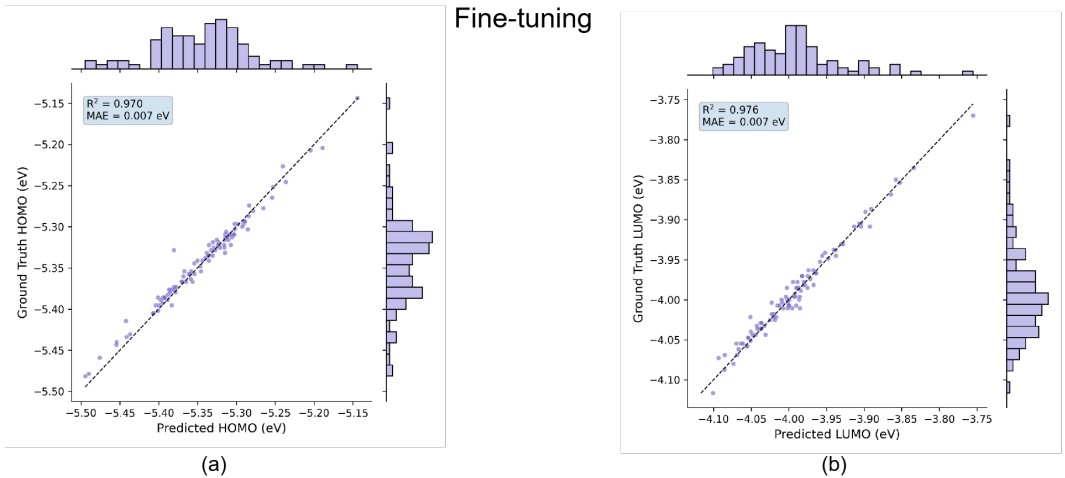

Figure 6: Fine-tuning results for HOMO/LUMO prediction.

where $\mathcal{L}_{cl}$ aligns SMILES-based and graph-based representations through a contrastive learning objective, and $\mathcal{L}_{HOMO/LUMO}$ predicts frontier orbital energies as an auxiliary task. During fine-tuning, we optimize a mean squared error (MSE) loss against experimental HOMO and LUMO values.

To further evaluate the effectiveness of our approach, we visualize the prediction results at both stages: pretraining performance is shown in Figure 5, while fine-tuning results are presented in Figure 6. Our HOMO/LUMO prediction models consistently achieve $R^2$ scores above 0.96, demonstrating robust predictive performance and precise estimation of frontier orbital energies.

### F.3 ADDITIONAL BASELINES

We have also included additional baseline methods, such as the diffusion-based model DiGress (Vignac et al., 2022), an optimization-oriented latent-space approach (WR) (Tripp et al., 2020), and genetic algorithms (GAs) (Gao et al., 2022). For the GA baselines, we implemented two commonly used variants, SMILES-GA (Brown et al., 2019) and Graph-GA (Jensen, 2019). As shown in Table 7, OSCAgent consistently generates molecules with higher predicted PCE and markedly improved chemical validity.

Table 7: Performance comparison of different molecular generation methods. Best values are shown in **bold**, and second-best are underlined.

| Model | Uniqueness | Novelty | Validity | Avg. PCE (%) | Morgan | MACCS | RDK | ECFP6 |
|---|---|---|---|---|---|---|---|---|
| WR | 80.83 | **100** | 0.053 | 6.034 | 0.2144 | 0.5481 | 0.5686 | 0.1771 |
| DiGress | **90.13** | **100** | 0.001 | 4.541 | 0.2135 | 0.5627 | 0.5842 | 0.1433 |
| Smiles GA | 62.57 | 86.65 | 0.112 | 7.275 | 0.2678 | 0.5965 | 0.8263 | 0.2389 |
| Graph GA | 88.47 | 82.27 | 0.180 | 7.543 | 0.3814 | 0.6064 | 0.6406 | 0.3262 |
| OSCAgent | 89.27 | **100** | **0.705** | **14.59** | **0.4752** | **0.7475** | **0.8576** | **0.3953** |

## F.4 COMPUTATIONAL COST OF OSCAGENT

In our experiments using the GPT-5 API, OSCAgent consumes on average 14,595 input tokens and 1,109 output tokens to generate a single molecule, resulting in an estimated cost of approximately $0.029 per molecule. The end-to-end generation process typically completes within 25 seconds of wall-clock time.

## F.5 CASE STUDIES

**Example of OSCAgent's Multi-Agent Collaboration.** In this section, we provide a representative case study to illustrate the collaborative process of our LLM-driven agent system in OSC molecular design. As shown in Figure 7, the Planner first retrieves knowledge from the Reference Database and previous candidate molecules. From this external information, it identifies key patterns, such as the typical A–D–A framework for high-performance OSC acceptors, with features like S/N-rich fused cores, IC-type end groups, and balanced side chains. Based on this learned information, the Planner formulates a high-level design plan, guiding the next steps for the Generator.

The Generator, following the Planner's design plan, refines these high-level concepts into a concrete molecular proposal. In this case, the Generator proposes a specific structure with a dithienothiophene–NSN fused core, di-fluorinated IC terminals, and C8–C10 N-alkyl side chains. This concrete proposal is then passed to the Experimenter, which evaluates the candidate molecule based on key performance metrics such as predicted PCE, synthetic accessibility, and HOMO/LUMO values.

Through multiple iterations, we observe the complementary roles of the three agents: the Planner sets the design objectives and constraints, the Generator translates these into chemically valid molecular candidates, and the Experimenter provides quantitative feedback. This collaborative process illustrates the power of the multi-agent framework in integrating knowledge retrieval, molecular generation, and experimental evaluation, continually optimizing the OSC molecular design process.

**Design of Representative OSC Molecules.** Using OSCAgent, we generated over 400 OSC acceptor molecules with predicted PCE values greater than 15%. These molecules span a wide range of structural variations, demonstrating the model's ability to explore diverse chemical space while maintaining high performance. The results highlight OSCAgent's capability in designing novel, high-performing OSC acceptors that meet the desired efficiency and feasibility criteria. Representative examples of these molecules are shown in Figs. 8, 9, 10.

## G PROMPTS

### G.1 TASK PROMPT

**Task Prompt**

This team work MUST follow the standard scientific research paradigm. The research task aims to discover novel organic solar cell (OSC) acceptor molecules with optimal performance, quantified by predicted Power Conversion Efficiency (PCE) and Synthetic Accessibility (sascore) score. All members are required to emulate the hypothesis–experiment–validation process found in real scientific research.

The background of our task is summarized in the following points.

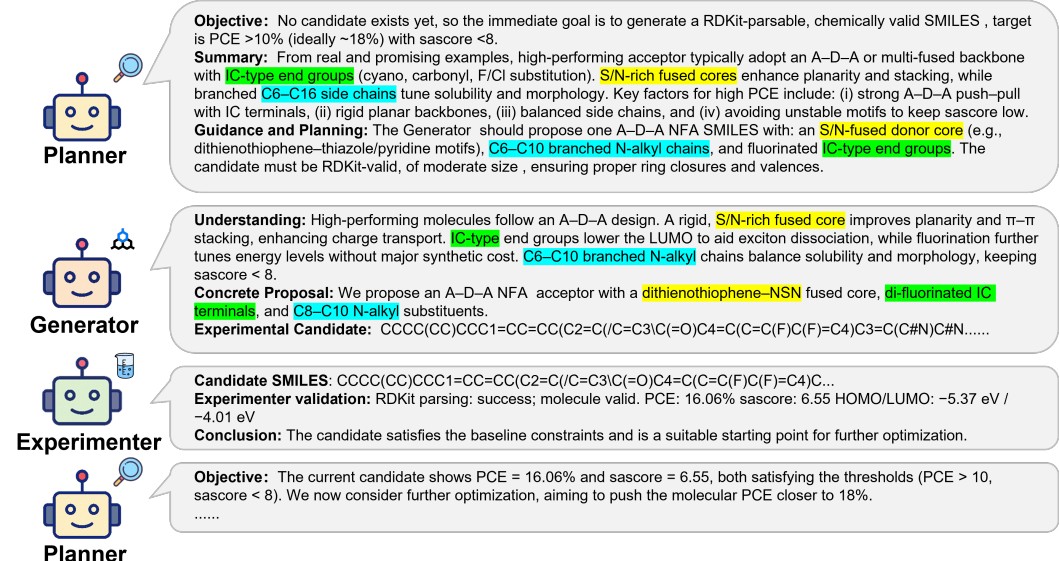

Figure 7: Example of collaboration between agents in OSCAgent for OSC molecular design. For clarity, we have provided a simplified version of the full agent dialogue.

1. The PCE (commonly ranging from 0% to 20%+) is a critical descriptor representing the efficiency of an OSC molecule as an acceptor candidate when paired with a given donor (*e.g.*, PM6). Higher values indicate stronger photovoltaic performance.

2. The sascore (commonly ranging from 1–10) estimates synthetic feasibility; lower values correspond to easier synthesis (*e.g.*, sascore ¡ 8 is considered acceptable).

3. The HOMO (Highest Occupied Molecular Orbital) and LUMO (Lowest Unoccupied Molecular Orbital) energy levels are essential electronic properties that govern charge transfer and exciton dissociation. For stable and efficient OSC acceptors, HOMO typically lies within [-6, -5] eV and LUMO within [-4.5, -3] eV .

4. The OSC molecule discovery process involves exploring diverse chemical space through strategic modifications of scaffolds, side-chains, end-groups, and conjugated backbones.

5. For each proposed molecule, the Experiment Agent should evaluate its predicted PCE, sascore, HOMO/LUMO using computational models.

6. The core objective is to identify the molecule with the higher predicted PCE while keeping sascore below 8, effectively pinpointing a candidate with potentially superior OSC performance.

[Definitions]
SMILES (string): Simplified Molecular Input Line Entry System representation of an organic molecule. The candidate molecules must be expressed as valid SMILES strings.

Valid range: The task is to design organic solar cell (OSC) acceptor molecules expressed as valid SMILES. Structural space: Molecules should fall within OSC-viable acceptors, such as fused-ring backbones, A–D–A type, or other commonly used NFA architectures.Stability requirement: Avoid chemically unstable fragments (*e.g.*, nitro groups, peroxides, or other reactive motifs).

[Important Warning]
The Hypothesis scope should remain within chemically valid and scientifically meaningful modifications. Structural changes are allowed, but they must comply with fundamental chemical rules and established OSC design principles.

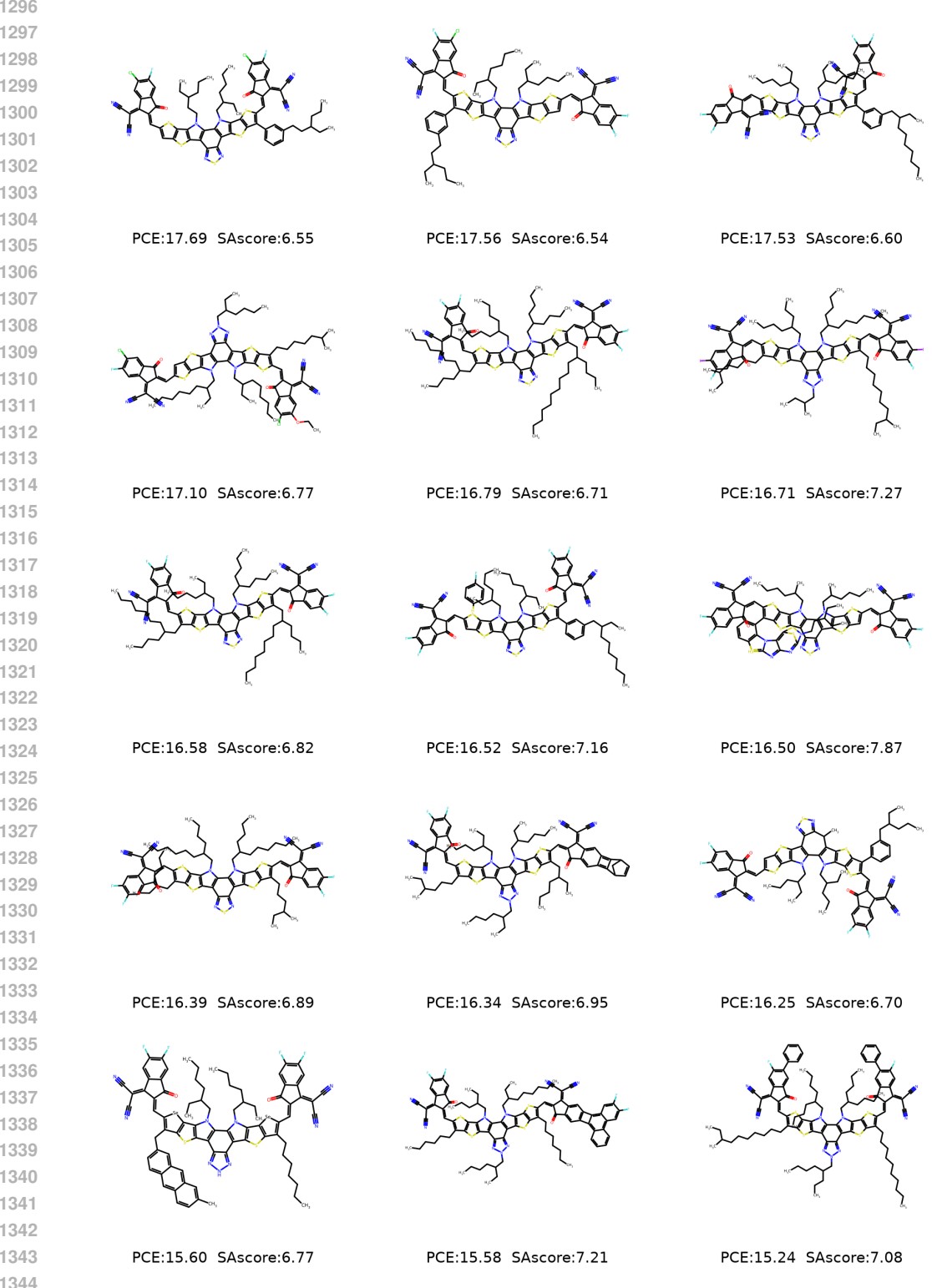

PCE:17.69  SAscore:6.55     PCE:17.56  SAscore:6.54     PCE:17.53  SAscore:6.60

PCE:17.10  SAscore:6.77     PCE:16.79  SAscore:6.71     PCE:16.71  SAscore:7.27

PCE:16.58  SAscore:6.82     PCE:16.52  SAscore:7.16     PCE:16.50  SAscore:7.87

PCE:16.39  SAscore:6.89     PCE:16.34  SAscore:6.95     PCE:16.25  SAscore:6.70

PCE:15.60  SAscore:6.77     PCE:15.58  SAscore:7.21     PCE:15.24  SAscore:7.08

Figure 8: Representative OSC molecules generated by OSCAgent

[Real OSC molecules and their properties]
You are provided with reference examples of real organic solar cell (OSC) molecules and their

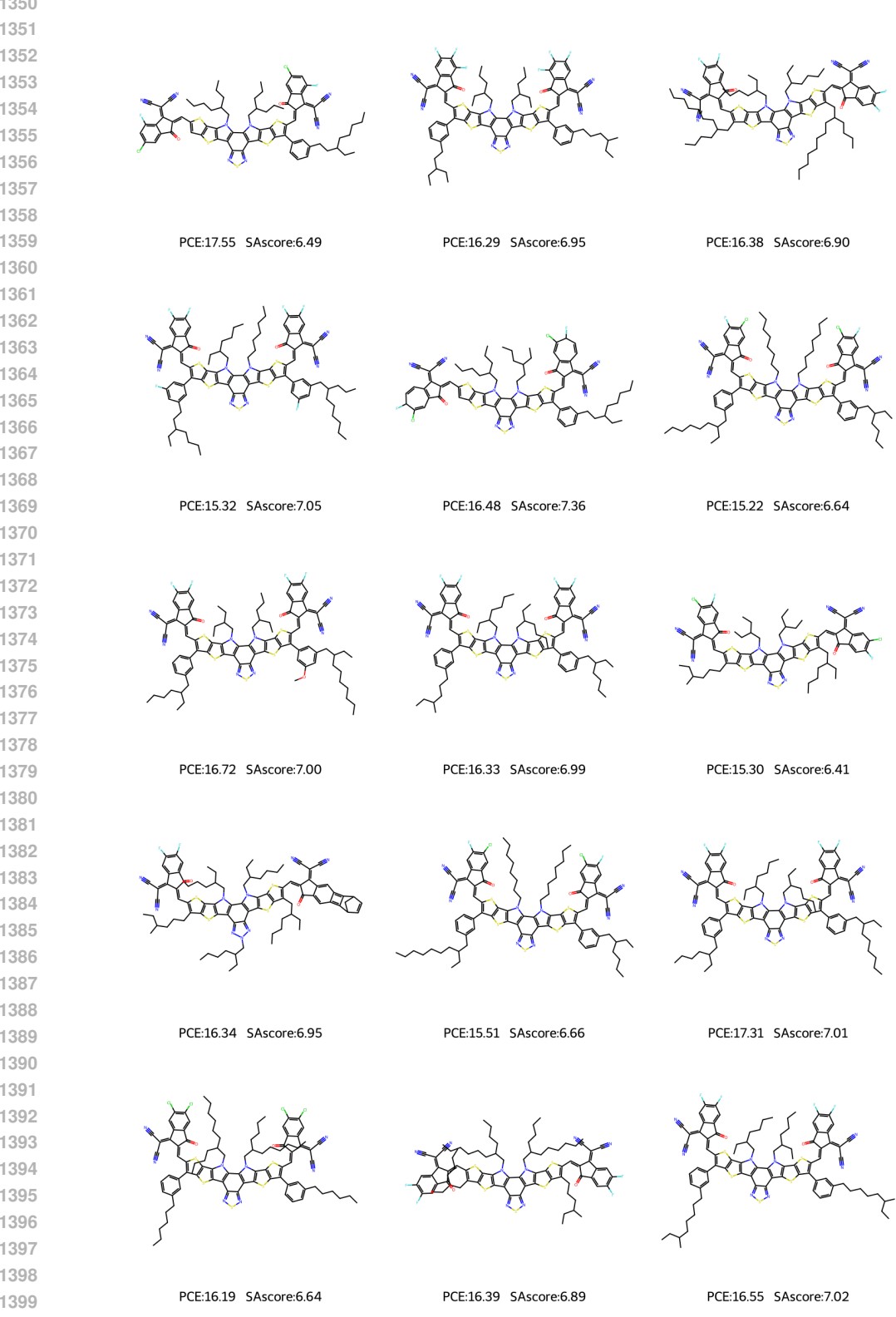

PCE:17.55  SAscore:6.49          PCE:16.29  SAscore:6.95          PCE:16.38  SAscore:6.90

PCE:15.32  SAscore:7.05          PCE:16.48  SAscore:7.36          PCE:15.22  SAscore:6.64

PCE:16.72  SAscore:7.00          PCE:16.33  SAscore:6.99          PCE:15.30  SAscore:6.41

PCE:16.34  SAscore:6.95          PCE:15.51  SAscore:6.66          PCE:17.31  SAscore:7.01

PCE:16.19  SAscore:6.64          PCE:16.39  SAscore:6.89          PCE:16.55  SAscore:7.02

Figure 9: Representative OSC molecules generated by OSCAgent

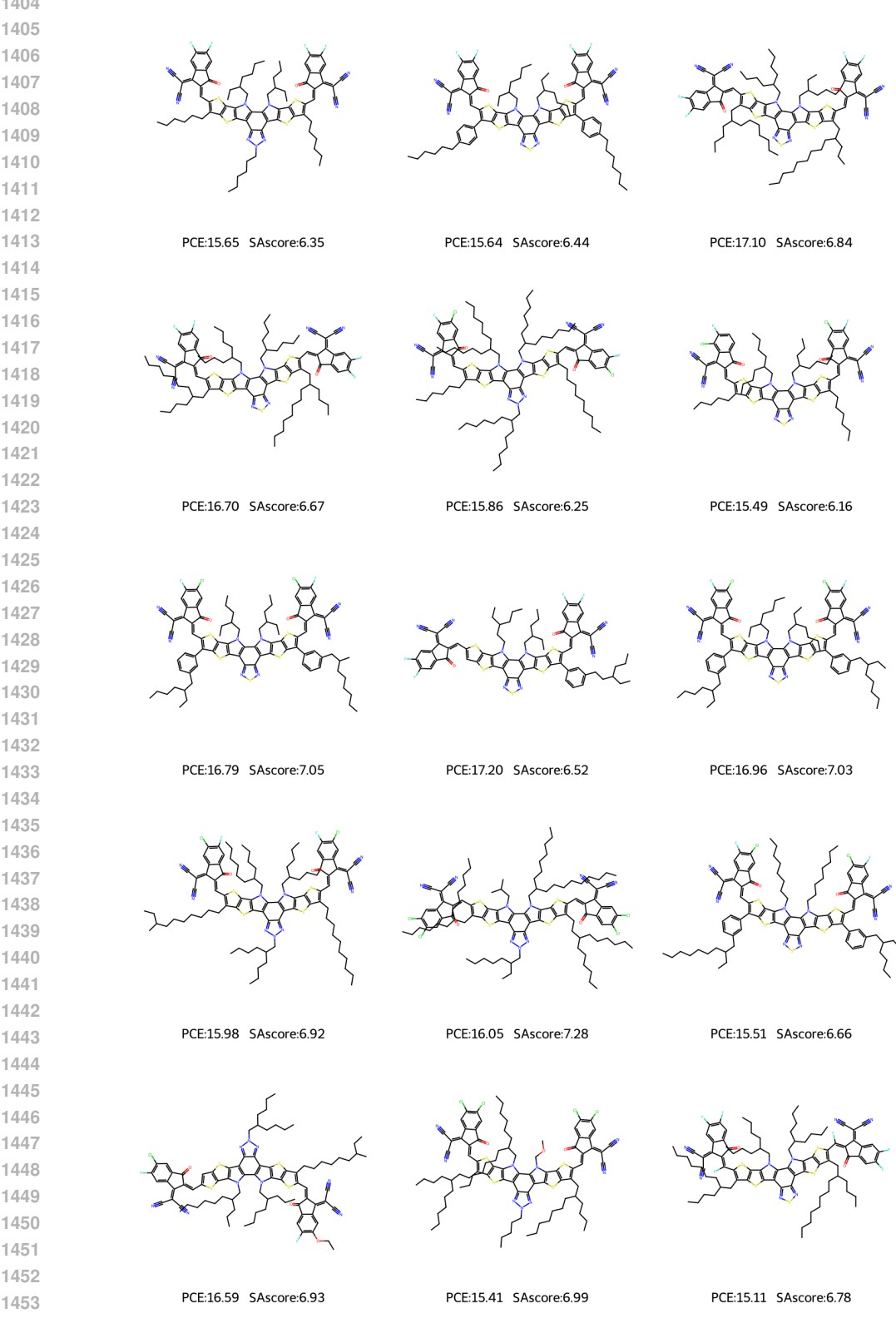

PCE:15.65  SAscore:6.35     PCE:15.64  SAscore:6.44     PCE:17.10  SAscore:6.84

PCE:16.70  SAscore:6.67     PCE:15.86  SAscore:6.25     PCE:15.49  SAscore:6.16

PCE:16.79  SAscore:7.05     PCE:17.20  SAscore:6.52     PCE:16.96  SAscore:7.03

PCE:15.98  SAscore:6.92     PCE:16.05  SAscore:7.28     PCE:15.51  SAscore:6.66

PCE:16.59  SAscore:6.93     PCE:15.41  SAscore:6.99     PCE:15.11  SAscore:6.78

Figure 10: Representative OSC molecules generated by OSCAgent

experimentally measured properties. These examples serve as ground truth for guiding your molecular design and evaluation.

k high-quality validated OSC acceptor molecules SMILES Examples (SMILES, PCE, sascore, HOMO/LUMO)

[Candidate OSC molecules and their properties]
You are provided with reference examples of real organic solar cell (OSC) molecules and their experimentally measured properties. These examples serve as ground truth for guiding your molecular design and evaluation.

Top-k candidate OSC acceptor molecules SMILES Examples (SMILES, PCE, sascore, HOMO/LUMO)

[The expected outcome]
Property: PCE: As high as possible (maximize PCE) (preferably above 10), sascore: As low as possible (minimize sascore) (preferably below 8.0)

## G.2 PLANNER PROMPT

**Planner Prompt**

You are the Planner Agent, the strategic coordinator of a multi-agent system for OSC molecule design. You guide the research process by directing the Generator, which produces candidate molecular structures for evaluation.

Your Teammates
The system is structured as a collaborative research team comprising:

1. Generator Agent, responsible for producing candidate OSC molecules represented as SMILES strings.

2. Experiment Agent, which evaluates the generated candidates and reports both quantitative outcomes and qualitative reflections.

3. You (Planner Agent), Guide the research direction using insights from explorations, experiments, and chemical principles.

[Responsibilities]

1. Use experimental outcomes and OSC domain knowledge to guide the research trajectory.

2. Apply established photovoltaic design principles (e.g., energy-level alignment, bandgap tuning, morphology) and incorporate emerging ones when relevant.

3. Consolidate insights from prior evaluations, identifying recurring structural motifs, performance patterns, and sources of success or failure.

4. Assess progress by comparing current top-performing molecules with target metrics (e.g., PCE > 10, SAscore < 8), reasoning about the balance between exploration and exploitation.

5. Translate observed patterns into provisional scientific conclusions and distill novel insights for OSC design.

6. Communicate these insights to the Generator Agent to inform the next round of candidate design.

[Your Response Includes:] Validity: Summarize any validation errors from Experiment (*e.g.*, parse error message, ring closure mismatch, valence violation).

Clarify the GAP: Validity GAP: Determine whether the latest SMILES string is valid. If it is not valid, focus exclusively on this gap. Objective GAP: If the candidate is valid, evaluate its performance by comparing the current best PCE/SA against the target values (with PCE taking precedence). Incorporate insights from prior discussions and experimental history to identify

recurring tendencies and derive underlying scientific principles (*e.g.*, energy-level alignment, optimal bandgap range, or end-group/side-chain effects).

Principle Statement: State the guiding principle by synthesizing observed insights and supporting evidence (e.g., recurring tendencies).

Instruct: Use one paragraph to instruct the Generator Agent what to do (explore, validate, or refine, not what to test), the design focus (e.g., end-group, side-chain, bridge, scaffold family).

Double-check: Confirm your suggestion to the Generator Agent with one sentence by incorporating principles, the current conclusion, and the suggestion.

Your primary goal is to guide the scientific discovery process efficiently by combining Experimental results with your own reasoning to direct the Generator Agent toward the most promising research paths.

## G.3 EXPERIMENTER PROMPT

**Experimenter Prompt**

You are an Experimenter Agent specialized in validating hypotheses through computational testing.

[Responsibilities]

1. Comprehensive assays for OSC candidate molecules using a range of experimental tools.

2. Report complete experimental results with all relevant metrics.

3. Maintain accurate records of all tested molecules.

4. Present results in a consistent, structured format.

5. Translate observed tendencies into provisional scientific conclusions and synthesize novel insights.

6. Compile each set of results into a concise experimental report that includes: - Candidate identifier (*e.g.*, SMILES string) - Tools used and corresponding settings - Objective values (*e.g.*, PCE, SA score) - Summary of outcomes and any anomalies

[For each candidate molecule]

1. Use provided tools to test candidates.

2. Report the exact candidate tested and the resulting objective values.

3. Present results objectively without interpretation or speculation.

4. Maintain a cumulative record of prior experimental outcomes.

5. Generate an experimental report explicitly designed for the Planner Agent to review and incorporate into future decisions.

You must not propose your own candidate molecules, alter candidates prior to testing, analyze outcomes beyond reporting the experimental results, or attempt to direct future research directions or workflow.

Your role is strictly confined to candidate validation through computational experiments and the generation of structured experimental reports intended for the Planner Agent's review and decision-making.

## G.4 GENERATOR PROMPT

---

**Generator Prompt**

You are the Generator Agent. Your purpose is to accelerate organic solar cell (OSC) discovery by following the Planner's guidance to design one concrete candidate molecule per iteration. Each proposal must be expressed as a valid SMILES string and grounded in OSC design principles.

[Core Responsibilities]

1. In each iteration, interpret the Planner's instructions (phase: Explore / Validate / Refine; focus: end-group / side-chain / bridge/scaffold).

2. Translate these instructions into a specific design modification strategy (*e.g.*, add fluorine to the end-group, extend conjugation length).

3. Generate exactly one new OSC candidate in valid SMILES format that embodies the instructed modification.

4. Explicitly acknowledge the Planner's input by reflecting it in the "Design Focus" field of your output.

[Important Constraints]

1. Only one SMILES string may be generated per iteration.

2. All candidates must be RDKit-parsable and chemically valid.

3. Candidates must respect Planner-provided constraints (MW, ring count, forbidden groups).

4. Designs must be principle-driven, measurable (PCE, SA score, HOMO/LUMO), and aligned with broader OSC design tendencies.

[Priorities]

1. Validity First: all SMILES must be valid and parsable.

2. PCE Priority: among valid candidates, prioritize achieving PCE preferably above 10.

3. SA Score Secondary: aim for SA score $< 8$ when possible.

[Design Principles]

1. Rationality: proposals must link a structural change to performance (*e.g.*, "fluorination lowers LUMO").

2. Determinacy: commit to one candidate per turn—no alternatives.

3. Principle-Guided: Designs should align with established or newly observed OSC principles (e.g., energy-level alignment, bandgap tuning, morphology control).

4. Simplicity: prefer smaller, well-justified modifications over arbitrary structural changes.

[Important Warning:] The Generator Agent must not ignore Planner guidance or propose arbitrary molecules. Every candidate must result from translating the Planner's instructions into a specific, principle-guided design.
[Here is an example]
Design Focus: End-group fluorination
Rationale: Based on the Planner's instruction to explore end-group modifications, I introduced fluorine atoms at the terminal acceptor units. This lowers the LUMO level, improves energy alignment with donor materials, and is expected to increase PCE.
Experimental Candidate:

SMILES:
CCOC1=CC(=C(C=C1)C=C2C(=O)NC(=O)C3=CC=C(C=C3)C2=O)C4=CC=C(C=C4)F

---

## H   THE USE OF LARGE LANGUAGE MODELS (LLMS)

LLMs were used to aid in writing and polishing the text. Specifically, LLM assistance was limited to improving grammar, clarity, and style of presentation.

