# OpenReview forum: "OSCAgent: Closing the Loop in Organic Solar Cell Discovery with LLM Agents"
_ICLR.cc/2026/Conference — ICLR 2026 Conference Withdrawn Submission_

### Official Review · Reviewer_F5D9 · 2025-10-21

**Soundness:** 2
**Presentation:** 3
**Contribution:** 3
**Rating:** 6
**Confidence:** 3

**Summary:**

This paper introduces OSCAgent, a multi-agent system for the discovery of organic solar cells (OSCs).

OSCAgent consists of three agents working in a loop:  (1) the planner, (2) the generator, and (3) the experimenter. The planner analyses existing designs (either from previous iterations of the loop or from external sources) and comes up with a research plan. The generator executes this plan to suggest molecules, which the experimenter then tests and provides the results back to the planner (through a report and updating a candidate database). The agents have access to different external tools (e.g., a power conversion efficiency prediction model, or RDKit) -- see Section 4.2.

In order to score designs in this loop, the authors develop their own power conversion efficiency (PCE) model. This is a multi-modal model, taking in graph, fingerprint, and SMILES representations of molecules and outputting a parameterized Gaussian distribution over the molecule's PCE score. The authors train this model in a two-step process, first on a related LUMO and contrastive learning prediction task, before then adapting the model for PCE regression with an uncertainty-aware loss (see Figure 2).

The experiments evaluate both (a) the OSCAgent (in terms of how well it can discover better OSCs) and (b) the PCE model. OSCAgent is compared against traditional ML and other sampling methods for suggesting better OSCs as well as single-step language models, which it is shown to substantially outperform (Table 1). The PCE model is compared against a series of other GNN, language, and traditional approaches where it also obtains better regression performance on an in-distribution test set. The authors perform some ablations on their model to judge the performance of various design decisions.

**Strengths:**

### S1 Overall high-level approach well presented
I thought overall the authors did a good job with the presentation of the high-level approach. Figures 1 and 2 are helpful for explaining the different parts, which I might have otherwise found confusing from the text alone (for instance, the PCE model has a somewhat complicated two-step training routine, and Figure 2 is useful in explaining which modules are frozen or trained in each stage of this routine and what loss is used). The method the authors propose is somewhat complicated with several different components, which means not everything can be explained in detail in the main paper due to space constraints. In the main though, I found the appendix useful for filling in these details (e.g., Section A contains detailed equations for each evaluation metric, Algorithm 1 describes how the algorithm for the database retrieval, or RAG, system works, etc).



### S2 Approach is general and can be applied to multiple domains
As the authors themselves mention in their conclusion, the approach seems broadly applicable to other molecular design tasks. Also, by creating three agents following the general scientific method of (1) hypothesize, (2) experiment, and (3) analyze results, the approach seems likely to also be relevant to other domains with some small modifications.

Having said that, to help me judge the approach’s significance here (plus the method's originality), it would have been nice to have seen in the discussion (or experiments) an idea of why this tri-agent approach is a lot better than previous multi-agent, "co-scientist" approaches such as those listed in the related work section. (Currently the related work section mostly just lists other agentic approaches rather than explaining how the presented approach differs from these previous methods, apart from being on a different application domain).

**Weaknesses:**

### W1 Some design choices seem somewhat arbitrary
Some of the design decisions of the multi-agent system seem fairly arbitrary. While the authors do a good job in running ablations on some of these choices (e.g., the retrieval augmentation strategy in Table 2), some of the design choices feel less well explored (e.g., where did the functional form for the candidate database retrieval scorer come from; why pretrain the PCE model on both a contrastive and LUMO loss; why split up the planning and generation agents). This means I find it hard to know which aspects of the approach are actually important.


### W2 The baselines evaluated against often seem quite weak
Often the baselines compared against seem quite weak, which makes it hard to just the significance of the proposed approach.

For instance, in table 1 my understanding is that no iterative approaches that can collect feedback on their initial suggestions and suggest new molecules (like OSCAgent can) were compared against? How well does a genetic algorithm-like approach work (for instance the Cao et al., 2025 or Greenstein & Hutchison, 2023 methods cited)? How about methods that adapt VAEs to optimization tasks like: Tripp, A., Daxberger, E., and Hernández-Lobato, J. M. Sample-efficient optimization in the latent space of deep generative models via weighted retraining. Advances in Neural Information Processing Systems, 2020.

Likewise in Table 3, it seems that the main improvement of the proposed PCE model comes from the handcrafted features used (which includes MACCS, RDK fingerprints, and others), but these are not provided to the baselines compared against?


### W3 Still not clear if the PCE model holds up well when optimized against
A common pitfall for optimizing against an ML scorer is that you end up exploiting the ML scorer model, rather than finding solutions to the underlying task that you are trying to solve. The authors mention the reliability of their scorer (line 178, 256), but this does not seem to be experimentally validated (Table 3 seems to be an in-distribution test set?). Is there a way to test how well the proposed model extrapolates? Perhaps by creating an out-of-distribution split or comparing against an independently trained model? (Extensions to integrating more realistic oracles, such as wet-lab experiments that would also solve this problem, are mentioned in the conclusion but are currently unexplored).

**Review scores above.**
These issues I mention with the baselines and evaluations are the main reason I have gone with a lower soundness score for now. Although, I still think the other advantages of the paper make me lean towards acceptance, I would like to see this addressed in the rebuttal.

**Questions:**

1. Figure 7 in the appendix was helpful to me for understanding what the conversation between the different agents look like. However, I was left uncertain with how the RAG in the planner worked. When does this occur? Is it run on the first step or only on the second and later iterations? Did you try different values of K for the different context sets and how were the values of 3 and 5 that were used for K arrived at?

2. The evaluation of the PCE model in table 3 seems to be mostly focused on how good a predictor the mean of the output Gaussian distribution is (by looking at the R^2 and MAE values). What about the predicted variance -- was this well calibrated? Do you have good evidence of the agents effectively using this model's uncertainty in particular?

3. The molecules suggested in Figure 3 actually seem to have somewhat large SAScores. How do these scores compare to those in the experimental datasets used (which presumably can be synthesized)? It seems that SAscore is treated as a constraint rather than an objective (line 1421). Did you try setting this constraint at a lower value than 8?

4. I'm surprised that the method still works so well without the experimenter and its subsequent feedback about the properties of the molecules suggested (although the molecular quality of this ablation goes down in Table 2, it seems to still perform a lot better than all of the competing approaches in Table 1). Is there any intuition for what allows the method to do so? (i.e., naively I might have expected it to be much more similar to the few-shot approach which as I understand it, also does not have access to feedback on its suggestions).

5. In table 1, how do the baselines compare in terms of just synthetic accessibility?


**A note on reproducibility**
The authors have a detailed appendix that lays out the prompts the different agents use as well as the hyperparameters the PCE model uses (Table 4). However, I would still find it very hard to reproduce the results of this paper on these alone. I hope the authors are able to release code (line 27 suggests this will happen) and provide the snapshots of the LLM models that were used.

---

> ### Author Response · Authors · 2025-11-21
> **Response to Reviewer F5D9 （1/5）**
>
> We sincerely thank the reviewer for the valuable comments and constructive feedback. We greatly appreciate the positive assessment of our high level presentation (S1), and we are grateful for the recognition of the generality and potential of our agent architecture (S2). Below, we provide detailed responses to the reviewer’s questions and concerns.
>
> ### Weaknesses1： Response to the reviewer’s concern on design details
>
> We thank the reviewer for the thoughtful comments regarding the clarity of several design choices. Below we provide detailed explanations for each component to make the motivations behind our system design more explicit.
>
> **1、Scoring function used in the Candidate Database.**
>
> The functional form is directly motivated by established OSC design principles。PCE (Power Conversion Efficiency) measures the performance of an OSC acceptor and therefore serves as the primary optimization objective in our discovery framework. SAscore reflects the difficulty of synthesizing a molecule and, in our scoring function, is used to penalize candidates that are hard to synthesize. The HOMO/LUMO feasibility term is derived from well established experimental energy-alignment windows; it encourages generated molecules to fall within the empirically observed orbital ranges of real OSC acceptors.
>
> Empirically, we find that removing all terms except the PCE component(optimizing solely for predicted PCE) leads to undesirable behavior. In this setting, the average SAscore of generated molecules increases from 6.42 to 6.54, indicating that the molecules become harder to synthesize. In addition, the distributional similarity to real OSC molecules also decreases
> | Model     |  Morgan  | MACCS  | RDK    | ECFP6  |
> |-----------|---------|--------|--------|--------|
> | Full   | 0.4752  | 0.7475 | 0.8576 |0.3953 |
> | Only PCE| 0.4272  | 0.7462 |0.8449| 0.3278 |
>
> **2、 Pretraining the PCE model with contrastive learning and an auxiliary LUMO prediction task.**
>
> Because experimentally measured PCE data are extremely limited, we pretrain the PCE model on the large Lopez dataset to acquire generalizable molecular representations before fine-tuning on OSC acceptors. To capture information at multiple levels, we adopt a multimodal architecture: the graph encoder models local bonding topology relevant to charge transport, while the SMILES encoder captures global structural syntax and long-range molecular patterns. These two modalities provide complementary signals that are especially valuable in low-data regimes.
>
> We use contrastive learning to align the graph and SMILES representations of the same molecule, encouraging the model to learn chemically meaningful, modality-invariant features rather than relying on modality-specific artifacts.
>
> In addition, the Lopez dataset provides tens of thousands of reliably computed DFT LUMO values, whereas experimental PCE measurements exist for only 1,027 molecules. LUMO energy is a key physical determinant of OSC performance—affecting electron affinity, exciton dissociation, and the achievable open-circuit voltage (VOC), as discussed in Sec. 3.2. Using LUMO as an auxiliary supervision signal therefore injects abundant and physically grounded information, helping the model learn electronic-structure features that are strongly correlated with downstream PCE.
>
> Empirically, we find that the auxiliary LUMO task leads to noticeably better downstream PCE prediction performance. When the auxiliary task is removed, the PCE predictor’s \(R^2\) drops from **0.713** to **0.671**, and the MAE increases from **1.686** to **1.794**.
>
> **3、why split up the planning and generation agents**
>
> We believe that these two stages require fundamentally different capabilities within OSCAgent. The Planner performs retrieval-augmented chemical reasoning: it analyzes high performance reference molecules, summarizes structural trends, integrates feedback from the Experimenter, and formulates targeted design strategies. This step is high-level, deliberative, and reasoning intensive. In contrast, the Generator focuses on low-level molecular construction—translating strategic instructions into chemically valid SMILES while respecting constraints such as syntactic validity, conjugation patterns, and OSC-specific motifs.
>
> For this reason, we separate the two roles. The Planner can conduct unconstrained reflective reasoning, while the Generator can concentrate solely on understanding the proposed strategies and producing molecules accordingly.

---

> ### Author Response · Authors · 2025-11-21
> **Response to Reviewer F5D9 （2/5）**
>
> ### Weaknesses2：Additional baselines and handcrafted features
>
> **1、Additional baselines**
>
> To address the reviewer’s concern regarding the choice of baselines, we have substantially expanded our experimental comparisons beyond those used in prior OSC molecular design studies.
>
> Most existing OSC design works rely almost exclusively on genetic algorithms (GAs), while diffusion-based generative models have not been systematically explored in this domain due to limited and highly specialized OSC datasets. To provide a more representative and competitive set of baselines, we included two widely used GA methods[4] (Smiles GA and Graph GA) and evaluated them directly against OSCAgent under identical scoring and evaluation conditions.
>
> To further strengthen the comparison, we incorporated a modern graph diffusion model, DiGress[1], as an additional SOTA generative baseline. DiGress was trained on the same molecular dataset used by DeepAcceptor[3] for its VAE baseline to ensure strict consistency and fairness across generative approaches.
>
> We also added an optimization-oriented latent-space method, the weighted-retraining (WR) strategy proposed by Tripp et al.[2]. WR iteratively updates a generative model by reweighting samples based on objective values, shifting probability mass toward high-scoring regions of chemical space. For fidelity, we trained WR using the authors’ official implementation.
>
> Across all these enhanced comparisons, which span GA-based approaches, diffusion models, and latent-space optimization methods, OSCAgent consistently generates molecules with higher predicted PCE, better synthetic accessibility, and markedly improved chemical validity.
>
> | Model     | Uniqueness | Novelty | Validity | Avg. PCE (%) | Morgan  | MACCS  | RDK    | ECFP6  |
> |-----------|------------|---------|----------|--------------|---------|--------|--------|--------|
> | WR    | 80.83      | **100**     | 0.053    | 6.034        | 0.2144  | 0.5481 | 0.5686 | 0.1771 |
> | DiGress   | **90.13**      | **100**     | 0.001    | 4.541        | 0.2135  | 0.5627 | 0.5842 | 0.1433 |
> | Smiles GA | 62.57      | 86.65   | 0.112    | 7.275        | 0.2678  | 0.5965 | 0.8263 | 0.2389 |
> | Graph GA  | 88.47      | 82.27   | 0.180    | 7.543        | 0.3814  | 0.6064 | 0.6406 | 0.3262 |
> | **OSCAgent**  | 89.27      | **100**     | **0.705**    | **14.59**        | **0.4752**  | **0.7475** | **0.8576** | **0.3953** |
>
> It is also important to note that the limited size and diversity of available OSC datasets pose intrinsic challenges for diffusion models, significantly restricting their expressive capacity and their ability to generate chemically meaningful OSC-specific structures. In contrast, OSCAgent combines domain knowledge, retrieval-augmented structural priors, and the reasoning capabilities of large language models with tool usage, enabling effective OSC molecular design even under data limited conditions and avoiding the low-validity issues observed in BRICS/VAE sampling.
>
> **2、Handcrafted features**
>
> Handcrafted features in our model refer to standard molecular fingerprint representations, which have been widely shown in prior molecular machine learning work to perform strongly on some small sized datasets[5][6]. Their good performance here is therefore expected and further supports the usefulness of incorporating chemically meaningful descriptors.
>
> Importantly, to ensure fairness, we did include a fingerprint-based baseline in Table 3. The Morgan + RF model—using Morgan fingerprints combined with a Random Forest—serves as a direct comparison based solely on handcrafted features. This baseline outperforms most deep learning models in Table 3, demonstrating both the strength of fingerprint-based representations and the necessity of including them in our hybrid feature design.
>
> Thus, the improvement of our PCE model does not arise from providing features unavailable to baselines, but rather from effectively integrating learned representations with chemically informative handcrafted descriptors.
>
> [1]DiGress: Discrete Denoising diffusion for graph generation, ICLR 2023
>
> [2]Sample-Efficient Optimization in the Latent Space of Deep Generative Models via Weighted Retraining, NIPS 2020
>
> [3]Accelerating the discovery of acceptor materials for organic solar cells by deep learning, npj computational materials
>
> [4]Sample Efficiency Matters: A Benchmark for Practical Molecular Optimization, NIPS 2022
>
> [5]MoleculeNet: A Benchmark for Molecular Machine Learning, Chemical science, 2018
>
> [6]Understanding the Limitations of Deep Models for Molecular property prediction: Insights and Solutions, NIPS 2023

---

> ### Author Response · Authors · 2025-11-21
> **Response to Reviewer F5D9 （3/5）**
>
> ### Weaknesses3：Reliability and extrapolation ability of the PCE prediction model
>
> To further address the reviewer’s concern regarding the reliability and robustness of our PCE predictor, we conducted additional experiments evaluating its extrapolation ability and cross-model consistency.
>
> **1、 Scaffold-based OOD evaluation.**
>
> To explicitly assess extrapolation, we constructed a scaffold-based out-of-distribution split using Bemis–Murcko scaffolds, ensuring that training and test sets share no scaffold overlap. As expected, performance decreases under this structural shift, but the degradation is relatively mild:
>
> - \(R^2\) decreases from **0.713** (standard split) to **0.663**
> - MAE increases from **1.686** to **1.801**
>
> This modest decline indicates that the model retains meaningful predictive power even on unseen scaffolds, suggesting that it is not simply memorizing the training distribution.
>
> **2、 Agreement with an independently trained model.**
>
> We also trained an independent Random Forest model using Morgan fingerprints as a separate PCE predictor. On the experimental test set, the Pearson correlation between our multimodal uncertainty-aware predictor and this RF baseline is **0.93**. For OSCAgent generated molecules—which are structurally more diverse and farther from the training distribution—the correlation remains high at **0.90**. This strong cross-model agreement between two architecturally distinct predictors indicates that top-ranked candidates are unlikely to be artifacts exploiting a single neural scorer.
>
> **3、Toward more realistic oracles.**
>
> We agree that incorporating more realistic scoring oracles, such as DFT calculations or wet-lab measurements, would further reduce the risk of model exploitation. As part of future work, we plan to integrate DFT-based recalibration and, ultimately, experimental feedback into the agent loop, enabling OSCAgent to operate with progressively more reliable and physically grounded evaluation signals.
>
> ### Questions1: RAG Mechanism and Choice of K
>
> **1、 How RAG is used in OSCAgent.**
>
> RAG is applied only in the first iteration of the planning process. The purpose of this initial retrieval step is to provide the Planner with a small set of high-performance reference molecules that anchor the search in a chemically meaningful region of the design space.
>
> In subsequent iterations, the Planner’s primary responsibility is to interpret and incorporate the Experimenter’s feedback. This feedback reflects the performance of molecules generated in earlier rounds. We think that introducing additional retrieval signals at this stage would mix external examples with internally generated evidence, which may confuse the Planner and interfere with its ability to optimize the search trajectory based solely on the feedback that accumulates within the iterative process.
>
> **2、 Selection of the K values.**
>
> We experimented with different retrieval sizes in the initial RAG step:
>
> Small K  (K = 2)
> Provided insufficient structural diversity and noticeably reduced distributional alignment between generated molecules and real OSC acceptors.
>
> Large K (K = 7)
> Although the performance remained stable, we found that increasing K beyond a moderate level did not lead to further improvement. Instead, it caused the prompts to become excessively long, especially because OSC molecules themselves are large and structurally complex, which in turn significantly increased token consumption without providing additional benefit.
>
> We therefore adopt an intermediate choice of \(K\), which provides the best balance between diversity and efficiency. In the Reference Database, we use \(K = 5\) to retrieve a diverse set of real high-performance OSC molecules. In the Candidate Database, we use \(K = 3\) to retrieve promising molecules that performed well in earlier runs. Real experimentally reported molecules play a more important role in this task, so a larger \(K\) is assigned to the Reference Database.
>
> | Model     |  Morgan  | MACCS  | RDK    | ECFP6  |
> |-----------|---------|--------|--------|--------|
> | K=(5,3)   | 0.4752  | 0.7475 | 0.8576 |0.3953 |
> | K=(7,7)| 0.4572  | 0.7481 |0.8587| 0.3792 |
> | K=(2,2)| 0.4216  | 0.7191 |0.8209| 0.3314 |

---

> ### Author Response · Authors · 2025-11-21
> **Response to Reviewer F5D9 （4/5）**
>
> ### Questions2: Uncertainty
> We thank the reviewer for raising the question regarding the variance estimates of our Gaussian PCE model. Although Table 3 focuses primarily on the quality of the predicted mean (R² and MAE), we have also evaluated the calibration and usefulness of the predicted aleatoric uncertainty.
>
> **1、Calibration of the predicted variance**
>
> We assessed uncertainty calibration using an area-based metric, obtaining an AUCE score of 0.1448. Considering the substantial experimental noise inherent in reported PCE values, this result indicates that the predicted variance generally aligns with the overall trend of true predictionerrors.Furthermore, the Pearson correlation between the predicted standard deviation and the absolute prediction error is 0.17, showing that higher predicted variance corresponds, on average, to higher empirical error. While not perfectly calibrated, this level of correlation is consistent with expectations given the noise in OSC measurements and the limited size of available datasets.
>
> **2、 How uncertainty is used within OSCAgent**
>
> In the current version of OSCAgent, the Experimenter records both the mean and standard deviation predicted by the Gaussian PCE model and returns them to the Planner. The uncertainty signal is used only qualitatively, providing contextual information that the Planner may consider during reasoning.The optimization logic and scoring functions rely primarily on the predicted mean and on chemically grounded constraints such as SAscore and HOMO/LUMO feasibility. This design choice avoids instability that could arise if the agent reacted too strongly to potentially noisy uncertainty estimates, which might otherwise degrade overall performance. Therefore, uncertainty is included as a reference signal rather than a direct optimization driver in the present implementation.
>
> ### Questions3 and Questions5: Synthetic accessibility score (SAScore)
>  **Comparison to experimentally reported OSC molecules**
>
> The average SAscore of OSCAgent-generated molecules is 6.42, which is very close to the average SAscore of 6.31 observed in experimentally synthesized high-performance acceptors. In real OSC datasets, SAscore and PCE show a moderate positive correlation of about 0.3, meaning that molecules with higher PCE values tend to be structurally more complex and consequently synthetically more challenging.
>
> For example, among experimentally reported acceptors with PCE > 15%, the average SAscore increases to 6.53, and real high-efficiency OSC acceptors can reach SAscore values as high as 7.84. These higher values reflect characteristics such as extended conjugation, fused-ring frameworks, and asymmetric end groups that naturally increase synthetic difficulty. Therefore, the SAscores observed in Figure 3 are well within the realistic and experimentally observed range for high-performance OSC acceptors.
>
> **Why SAscore is treated as a constraint rather than an objective.**
>
> Following  prior OSC works[3], we treat SAscore as a feasibility constraint, not as an optimization objective. Directly minimizing SAscore would bias the generator toward overly simple structures, which tend to be low-performing and lack the chemical richness found in real OSC acceptors.
>
> **Why we do not set a lower SAscore threshold.**
>
> Since real, synthetically feasible high-performance acceptors can have SAscore values up to 7.84, lowering the constraint below 8 would exclude many legitimate, experimentally viable motifs. This would unnecessarily shrink the chemical search space and hurt performance. Keeping the threshold at 8, consistent with prior literature, provides an appropriate balance between synthetic feasibility and realistic molecular diversity.
>
> **Baseline comparison in terms of SAscore**
>
> Many baseline methods tend to generate structurally simple conjugated fragments with much lower SAscores. Although these structures appear easier to synthesize, they generally lack the fused-ring cores, strong acceptor units, and extended π-systems needed for meaningful OSC function. Their lower SAscore reflects underspecified or under-functionalized structures, rather than realistic synthesizability.
>
> In contrast, OSCAgent produces molecules whose SAscores match the complexity level observed in high-performance real OSC acceptors, while also achieving substantially higher predicted PCE and much higher chemical validity.

---

> ### Author Response · Authors · 2025-11-21
> **Response to Reviewer F5D9 （5/5）**
>
> ### Questions4：Explanation of the performance of the “no-Experimenter” ablation
>
> The “no-Experimenter” ablation still retains all other components of the framework, including the RAG mechanism, carefully designed background prompts, and the multi-turn iterative interaction . Although the ablation removes access to quantitative feedback from the Experimenter, it does not eliminate the inductive biases and reasoning structure embedded in the system. The ablated variant outperforms other baselines for three main reasons:
>
> **1、RAG and prompt design anchor the Planner close to the real OSC design space**
>
> Without the Experimenter, the Planner can no longer validate its strategy quantitatively, which explains the noticeable degradation in molecular quality shown in Table 2.
> However, the combination of:retrieval augmentation based on high-performance OSC reference molecules, and carefully constructed background prompts, still guides the Planner’s reasoning toward realistic and chemically meaningful design patterns. As a result, even without experimental feedback, the generated molecules remain much closer to the empirical distribution of real OSC acceptors than those produced by baselines lacking such strong domain priors.
>
> **2、Decoupling the Planner and Generator improves controllability**
>
> Unlike few-shot prompting baselines, our architecture separates reasoning (Planner) from molecule synthesis (Generator). The Planner produces high-level structural guidance such as π-bridge choices, acceptor–donor motif balance, and conjugation length—that constrains the Generator’s search space.
>
> This division of roles reduces the likelihood that the Generator produces arbitrary or syntactically unstable SMILES strings and helps keep the generation process aligned with established OSC design principles. As a result, the system remains far more controllable and stable than single pass few-shot prompting, even in the absence of Experimenter feedback.
>
> **3、Iterative refinement persists even without the Experimenter**
>
> Although the Experimenter is removed in this ablation, the iterative multi-turn interaction between the Planner and Generator remains fully intact. The Planner continues to read, interpret, and qualitatively assess molecules generated in earlier rounds, and then refines its instructions accordingly. This allows the Planner to perform a form of self-guided refinement, steering the Generator toward more promising regions of chemical space based on internal reasoning and retrieved prior knowledge.
>
> In contrast, the few-shot baseline produces molecules in a single step, without any iterative reasoning or self-correction mechanism.

---

> > ### Comment · Reviewer_F5D9 · 2025-11-22
> > **Thanks to the authors for their rebuttal**
> >
> > Thank you to the authors for the rebuttal and following up on my comments and also the questions.
> >
> > Going through the original main points:
> >
> > **W1 Some design choices seemed arbitrary**.
> > Appreciate the additional comments here. While many of the choices don't seem unreasonable they do still seem complex and I worry they may just be overfitting to the data/setup (it's hard to know how many alternative reasonable ideas might have been tried). The new ablation for the scoring function (using only PCE) seems to show that the complex functional form of the scoring function for the candidate database only makes a small difference (the reduction in the SAscore is very minor).
> >
> > **W2 Additional baselines**.
> > Thank you a lot for running these new stronger baselines for the main agentic workflow which are iterative and so seem a fairer comparison to make. Great to have these as a point of comparison and encourgaing to see OSCAgent still does better.
> >
> > One point I am confused on with these though is the comment that says "OSCAgent consistently generates molecules with [...] better synthetic accessibility" which seems in conflict with the latter part of the rebuttal arguing "Many baseline methods tend to generate structurally simple conjugated fragments with much lower SAscores" and that actually very high synthetic accessibility is likely not actually a thing you want?
> >
> > Furthermore, appreciate the RF + fingerprint baseline that you initially included, but I think my point still stands: if including fingerprints is the main factor that seems important why not compare against more methods that include this information with learned representations e.g. by adding this information to the current baseline models, or by considering one of the many of the exisiting methods that work similarly, e.g.,
> > Analyzing Learned Molecular Representations for Property Prediction; J. Chem. Inf. Model. 2019, 59, 8, 3370–3388
> >
> > **W3 Optimizing against the PCE model**.
> > Thanks for your new experiments here; think the new OOD generalization results are helpful (as well as the ambition to consider more realistic oracles which seems reasonable to postpone to future work).
> >
> >
> > Thanks again for the rebuttal and will also follow the discussion with the other reviewers before updating my review!

---

> > > ### Author Response · Authors · 2025-11-25
> > >
> > > We thank the reviewer for the follow-up comments and for the opportunity to further clarify the remaining concerns. Below, we provide a more detailed explanation addressing the reviewer’s question.
> > >
> > >
> > > ### Weaknesses1：Some design choices seemed arbitrary
> > >
> > > **Question1：On the design of our scoring function**
> > > We thank the reviewer for raising this question. Our scoring function design is inspired by standard multi objective optimization practices commonly used in molecular design[1]. Under this framework, we assign higher scores to molecules that simultaneously exhibit high predicted PCE, reasonable synthetic accessibility, and proper HOMO–LUMO level alignment, so that the agent is guided toward chemically meaningful candidates.
> > >
> > > Our experiments confirm that this design is effective: the generated molecules exhibit PCE, synthetic accessibility, and energy-level characteristics that closely match the distribution of real high-performing OSC acceptors. This demonstrates that the scoring function helps maintain chemical plausibility while still enabling high-PCE molecular discovery.
> > >
> > > **Question2：Why is the numerical change in SAscore relatively small?**
> > >
> > > After removing the other terms in the scoring function and keeping only PCE, we do observe an increase in the average SAscore. However, the magnitude of this increase is relatively small. We believe this can be explained by the following two factors:
> > >
> > > 1、Our retrieval-augmented generation system uses both real molecules and generated candidates as references. Real OSC acceptors overwhelmingly have SAscore < 7, and our RAG module is intentionally designed to emphasize learning patterns from these real molecules. As a result, even when the SA term is removed from the scoring function, the Generator is still guided by realistic examples whose SA values lie in a narrow, chemically reasonable regime. This naturally mitigates the increase in SAscore.
> > >
> > > 2、Our data analysis shows that the vast majority of real, high-performing OSC acceptors fall within the 6–7 SA range, and the same is true for most molecules generated by OSCAgent. Because the SA distribution is already highly concentrated within this narrow interval, its overall mean has a limited dynamic range. Under such a constrained distribution, even an average increase of 0.1 is already meaningful.

---

> > > ### Author Response · Authors · 2025-11-25
> > >
> > > ### Weaknesses2：Additional baselines.
> > >
> > > **Question1：Clarification on “better synthetic accessibility”
> > >
> > > For OSC molecules, the primary design objective remains the PCE value. In our work, we consider SAscore not as an independent optimization target, but to ensure that OSC molecules with high predicted PCE also maintain a normal and realistic level of synthetic complexity.
> > >
> > > Because real OSC data show a mild positive correlation between PCE and SAscore, completely ignoring SAscore during generation may cause the model to pursue high PCE by producing overly complex or atypical structures that deviate from realistic OSC chemistry. Conversely, generating overly simple structures (with very low SA) would ignore the level of π-conjugation required for functional OSC molecules. Neither extreme is desirable.
> > >
> > > Our intention is thus: To generate OSC molecules with high PCE while keeping their synthetic accessibility within the normal range observed in real OSC acceptors.
> > >
> > > Experimentally, this is indeed what we observe:
> > > 1、The average SAscore of OSCAgent-generated molecules is close to that of real OSC molecules, indicating realistic structural complexity.
> > > 2、Most generated molecules fall in the SAscore range of 6–7, which is exactly the same range where real high-performing OSC acceptors lie.
> > > 3、Only around 1% of generated molecules have SAscore > 8, showing that OSCAgent rarely produces overly complex structures.
> > >
> > > Therefore, when we state that “OSCAgent generates molecules with better synthetic accessibility,” we do not mean “lower SA is better.”  Some baselines collapse to overly simple structures with abnormally low SA, which are unrealistic for high-performance OSCs.
> > >
> > > **Question2：Additional baselines about fingerprints**
> > >
> > > To further evaluate the effectiveness of molecular fingerprints for OSC property prediction, we additionally introduce three new baselines:
> > >
> > > 1、Morgan + MLP
> > > A simple multilayer perceptron trained solely on Morgan fingerprints derived from the molecules.
> > >
> > > 2、D-MPNN[2] (with fingerprints)
> > > A hybrid model that augments the D-MPNN representation with concatenated molecular fingerprint features for downstream prediction.
> > >
> > > 3、Our model (fingerprints only)
> > > A variant of OSCAgent’s predictive module where we remove graph and SMILES features and retain only fingerprint-based inputs.
> > >
> > > Across these baselines, we observe a consistent trend:
> > > fingerprint-based models generally outperform models that rely only on molecular graphs or SMILES representations, confirming that fingerprints remain a strong and informative feature space for this task.
> > >
> > > However, while models such as Morgan+MLP and D-MPNN (with fingerprints) do improve over graph-only or SMILES-only approaches, their performance still lags behind the simple RF baseline, suggesting that MLP-style predictors may not effectively capture the non-linear structure–property relationships present in OSC molecules.
> > >
> > > In contrast, our model using fingerprints alone outperforms all other fingerprint-based baselines, demonstrating that the mixture-of-experts (MoE) architecture we adopt is particularly effective at leveraging discrete feature representations. These results further highlight the flexibility and robustness of our predictive model across different molecular feature modalities.
> > >
> > >
> > > | Model     | R^2 | MAE |
> > > |-----------|------------|---------|
> > > | Morgan + RF    | 0.649     |1.875     |
> > > | Morgan + MLP  | 0.637      | 1.881     |
> > > | D-MPNN | 0.642      | 1.878   |
> > > | Our model (fingerprints)  | 0.658      | 1.867   |
> > > |
> > >
> > >
> > > [1]Junction Tree Variational Autoencoder for Molecular Graph Generation
> > > [2]Analyzing Learned Molecular Representations for Property Prediction

---

### Official Review · Reviewer_UQiv · 2025-10-28

**Soundness:** 1
**Presentation:** 2
**Contribution:** 2
**Rating:** 2
**Confidence:** 5

**Summary:**

The paper presents OSCAgent, a multi-agent framework for organic solar cell (OSC) molecular discovery. The system integrates three agents (Planner, Generator, and Experimenter) to enable an automated pipeline for OSC molecule design. The Planner retrieves knowledge from literature and past candidates, the Generator proposes new acceptor molecules, and the Experimenter evaluates their predicted properties and provides feedback for model improvement. The authors claim that OSCAgent can autonomously generate chemically valid and synthetically accessible OSC molecules with predicted efficiencies up to 18%, outperforming existing baselines.

**Strengths:**

* The concept of using an agentic AI system for accelerating OSC material discovery is timely and relevant, given the growing interest in applying LLMs and autonomous systems in materials science.

* The overall framework combining retrieval, generation, and evaluation in a closed-loop pipeline is an interesting direction that could inspire future developments.

**Weaknesses:**

* Technical and conceptual flaws: The fundamental formulation of the problem is questionable. Predicting PCE based solely on the acceptor molecule without specifying or assuming a donor counterpart is scientifically unsound. Since OSC efficiency depends critically on the donor–acceptor pair and their interfacial energy alignment, this limitation invalidates most results and conclusions.

* Molecule quality: Many generated molecules shown in Figure 8 appear chemically unrealistic. While they contain features of non-fullerene acceptors (NFAs), they lack the required symmetry, which is a key characteristic for ensuring proper molecular packing, balanced charge transport, and favorable optical/electronic properties in NFAs.

* PDF quality issues: The submitted PDF file has display errors, with several figure pages blank or unreadable.

Given the flawed problem setup and unconvincing experimental results, I recommend rejection.

***Minor comments:***

* The claim “Our work introduces an LLM-based multiagent framework that enables closed-loop discovery of novel OSC acceptors” is overstated. In-silico evaluation (using the Scharber model or SA score) cannot be considered closed-loop discovery. Real closed-loop discovery requires actual synthesis/testing and iterative improvement based on experimental feedback or human input.

* In Section 2 (Related Work), the discussion of “LLM Agents for Science” is disconnected from the proposed method. The paper would benefit from a clearer positioning of OSCAgent relative to existing LLM-based scientific agent systems.

* In Section 3.1 (Organic Solar Cells), while the focus on acceptor design is stated, the rationale for excluding donor molecules should be elaborated.

* Typographical issue in Figure 1 caption: “com- prehensive” → “comprehensive.”

**Questions:**

1. What donor material was assumed for PCE calculations? Since PCE depends on both donor and acceptor energy levels, absorption spectra and their miscibility in a specific solvent, this needs clarification.

2. Were the HOMO–LUMO levels computed using quantum chemistry simulations or taken from experimental data?

3. The paper mentions using the Lopez et al. (2017) dataset from the Harvard Clean Energy Project, which contains donor–π–acceptor (D–π–A) type molecules intended for *donor* design. How was this dataset used to improve acceptor generation, and how was domain mismatch addressed?

4. How does the framework handle conflicting evaluation signals from the Experimenter, e.g., high predicted PCE but poor energy level alignment (HOMO–LUMO mismatch)?

---

> ### Author Response · Authors · 2025-11-21
> **Response to Reviewer UQiv （1/4）**
>
> We sincerely thank the reviewer for the thoughtful evaluation of our submission and for the encouraging remarks regarding the timeliness and relevance of applying agentic AI systems to accelerate OSC material discovery. We also thank the reviewer for the constructive and insightful comments. Below, we provide detailed responses to the reviewer’s concerns.
>
>
> ### Weaknesses1 and Question1：About technical and conceptual flaws
>
> We appreciate the reviewer’s concern and fully agree that device-level OSC performance depends on donor–acceptor (D/A) interactions. Because our work focuses on identifying high-potential acceptor molecules, our objective is to model the relationship between an acceptor’s structure and its intrinsic PCE potential. We emphasize, however, that our formulation does not introduce an ad-hoc simplification; rather, it follows a well established and widely used labeling strategy in peer reviewed OSC machine learning research. We also clarify that OSCAgent does not completely ignore donor effects. We elaborate on this below.
>
>
>
> **1、 Our formulation strictly follows DeepAcceptor[1] (npj Computational Materials, 2024)**
>
> Like DeepAcceptor, our work focuses on identifying high-potential acceptor molecules. We adopt the same data curation protocol and use the same underlying dataset as DeepAcceptor. As stated in their paper:
>
> “To obtain a large training dataset and facilitate the discovery of high-performance acceptors, the type of donor was ignored. If a molecule appears several times in the literature, its maximum PCE is chosen.” — Sun et al., npj Computational Materials (2024)
>
> Following this widely used practice, we also assign to each acceptor the maximum reported PCE from the literature when training our PCE prediction model. This labeling strategy enables the deep learning model to capture the intrinsic performance potential of an acceptor molecule and supports the discovery of high-potential acceptor candidates.
>
> **2、This labeling strategy is commonly used across multiple peer-reviewed OSC ML studies (for both donors and acceptors)**
>
> Beyond DeepAcceptor, several published studies also adopt similar processing strategies, in which multiple entries corresponding to the same molecule are merged and the maximum reported PCE is used as the learning target to represent the potential of that OSC molecule. This indicates that such labeling approaches are widely used and broadly accepted within the OSC machine learning community.
>
> **Science Advances — Sun et al.[2]**
>
> In this study, all entries corresponding to the same donor molecule were merged, and the highest reported power conversion efficiency (PCE) was used as the target value. The PCE of the donor molecules was then predicted through machine learning. The authors explicitly state:
>
> “If a certain donor material has been reported several times, the highest PCE is chosen. All these criteria ensured the model can learn the maximum potential of a certain material.” - Sun et al., npj Science Advances (2019)
>
> Through independent modeling of donor molecules, their machine learning approach was able to identify new high potential donor candidates.
>
> **AAAI — Ding et al. (RingFormer)[3]**
>
> In this study, the authors proposed a new model architecture to predict the potential PCE of individual OSC molecules. Their dataset includes both donors and acceptors, and in both cases the authors adopt the same processing strategy. Specifically, the paper states:
>
> “Due to potential duplications arising from one donor polymer molecule working with different acceptors, resulting in different PCE values, molecules with the same nickname are merged. The largest PCE value is considered the ground truth.”
>
> and
>
> “During dataset curation, pairs of molecules with the same acceptor molecule are merged based on SMILES strings, and the largest PCE value is considered the ground truth.”
>
> Taken together, these studies indicate that using machine learning to predict the performance potential of individual OSC molecules is a well-established approach in prior OSC ML literature. Our methodology is fully consistent with these precedents and should not be regarded as scientifically unsound.
>
> [1]Accelerating the discovery of acceptor materials for organic solar cells by deep learning, npj computational materials 2024
>
> [2]Machine learning–assisted molecular design and efficiency prediction for high-performance organic photovoltaic materials, Science Advances 2019
>
> [3]RingFormer: A Ring-Enhanced Graph Transformer for Organic Solar Cell Property Prediction, AAAI 2024

---

> ### Author Response · Authors · 2025-11-21
> **Response to Reviewer UQiv （2/4）**
>
> **3. Our method does not ignore donor effects，LLMs naturally embed donor–acceptor knowledge**
>
> Although our predictive model and optimization objective are formulated at the acceptor level (consistent with DeepAcceptor), the LLM agents themselves are not blind to donor–acceptor interactions. Large language models pretrained on chemistry-related literature inherently capture general OSC knowledge. In practice, both the Planner and Generator frequently refer to donor–acceptor compatibility patterns, frontier-orbital alignment, and morphology-related considerations.
>
> From our agent-chain logs, we observe that the Planner explicitly reasons about compatibility with commonly used donors (e.g., PM6), indicating that donor-related information is naturally incorporated through the LLM’s pretrained knowledge.
>
> Thus, while the evaluation loop focuses on modeling acceptor properties, the LLM-driven reasoning process embeds donor-aware patterns implicitly learned during pretraining.
>
> **4. OSCAgent can be directly extended to donor-specific acceptor design when such datasets are available**
>
> If datasets containing PCE values for specific donor–acceptor pairings become available, OSCAgent can be adapted to design acceptors conditioned on a fixed donor with minimal changes. When the donor is predetermined, the PCE predictor only needs the acceptor features as input, since the donor information is constant for all samples. The Planner can retrieve examples relevant to that donor, and the Generator can be conditioned on the user-specified donor target during molecule creation. The main limitation of this extension is data availability datasets at the donor–acceptor pair level are typically much smaller, which may constrain model performance. We will describe this potential extension explicitly in the revised manuscript.

---

> ### Author Response · Authors · 2025-11-21
> **Response to Reviewer UQiv （3/4）**
>
> ### Weaknesses2：About molecule quality
> We sincerely thank the reviewer for the insightful comment. Below we address the concern regarding molecular symmetry.
>
> **1. Symmetry was common in early NFAs, but modern high-performance NFAs increasingly adopt asymmetric designs.**
>
> We acknowledge that many early non-fullerene acceptors  employed highly symmetric A–D–A motifs. However, symmetry is no longer a predominant design requirement. Recent advances in OSC materials clearly demonstrate that asymmetric acceptor architectures can provide substantial performance benefits. Representative examples include:
>
> Asymmetric electron acceptor enables highly luminescent organic solar cells with certified efficiency over 18% (Nature Communications, 2022)
>
> Non-fullerene acceptor with asymmetric structure and phenyl-substituted alkyl side chain for 20.2% efficiency organic solar cells （Nature energy, 2024）
>
> Asymmetric Non-Fullerene Small-Molecule Acceptors toward High-Performance Organic Solar Cells （ACS Central Science, 2021）
>
> Asymmetric non-fullerene acceptors with an imide-containing end group for high-performance organic solar cells （JMCC, 2023）
>
> Asymmetric Alkyl Chain Engineering of Non-fullerene Acceptors for Efficient Organic Solar Cells (CJPS, 2025)
>
> Recent advances and prospects of asymmetric non-fullerene small molecule acceptors for polymer solar cells （ J. Semicond, 2021）
>
> These and many other studies collectively highlight that asymmetry is now a widely explored and scientifically grounded direction for achieving higher OSC performance.
>
> **2. Prior computational OSC design studies also do not treat symmetry as a required chemical constraint.**
>
> Across previous computational OSC design works[1][5], symmetry has never been imposed as a chemical constraint or validity requirement. These studies routinely generate and evaluate asymmetric donor and acceptor structures, and their generative pipelines do not include any symmetry-related filtering. For example, works such as “High-throughput molecular design of donors and non-fullerene acceptors based on convolutional neural networks[4]” report numerous asymmetric candidate molecules without treating symmetry as a factor in molecular validity or performance assessment. Collectively, these studies show that structural symmetry is not considered essential in computational OSC molecular design.
>
> **3. We will provide additional molecular visualizations to ensure clarity.**
>
> We will provide additional molecular visualizations to ensure clarity.We understand the reviewer’s concern about the representativeness of the examples shown in Figure 8,9,10. In the revised manuscript, we will include more diverse and comprehensive molecular visualizations.
>
> [4] High-throughput molecular design of donors and non-fullerene acceptors based on convolutional neural networks, JCIM 2025
>
> [5] Molecular design of organic photovoltaic donors and non-fullerene acceptors: a combined machine learning and genetic algorithm approach, JMCC 2025
>
> ### Weaknesses3：About PDF quality issues
>
> After multiple rounds of verification on our side, we were unable to reproduce the PDF display issues—the figures render correctly on all machines and PDF viewers we tested (including Adobe Acrobat, Preview, and browser-based viewers). The issue may be related to a specific viewer, caching, or local rendering settings.
>
> If the reviewer still encounters display problems, we would greatly appreciate being informed of the affected pages or figures. If permitted, we are also happy to provide an additional copy of the full PDF or individual figure files to ensure smooth evaluation.

---

> ### Author Response · Authors · 2025-11-21
> **Response to Reviewer UQiv （4/4）**
>
> ### Minor comments 1：Clarification regarding the use of “closed-loop” in our paper
> We thank the reviewer for the constructive feedback. The term “closed-loop” in our manuscript follows the same usage as in recent agentic science frameworks (e.g., InternAgent[6]). In our work, the Planner, Generator, and Experimenter interact iteratively to refine candidate OSC molecules based on in-silico predictive feedback. At this stage, the loop does not involve real synthesis or experimental measurements.
>
> We will clarify this distinction in the revised manuscript and will incorporate the reviewer’s additional suggestions to improve the overall clarity and precision of the paper.
>
> ### Minor comments 2,3,4：Manuscript revisions
> We thank the reviewer for these helpful suggestions. We will revise the manuscript accordingly.
>
> First, regarding the discussion of “LLM Agents for Science” in Section 2, we agree that its connection to our method can be made clearer. In the revised version, we will explicitly position OSCAgent within the broader landscape of LLM-based scientific agent systems.
>
> Second, we will provide a clearer rationale for focusing on acceptor design rather than donor molecules. This formulation follows established practice in prior peer-reviewed OSC machine-learning studies (e.g., DeepAcceptor, RingFormer), which learn the intrinsic potential of acceptors using the maximum reported PCE. We will elaborate on why this labeling strategy is widely used and how it aligns with our goal of discovering high-potential acceptor molecules(Appendix H).
>
> Finally, we appreciate the reviewer’s note about the typographical issue in Figure 1, and we will correct “com-prehensive” to “comprehensive” in the revised manuscript.
> ### Question2：HOMO–LUMO data source
>
> The HOMO–LUMO level data used in our work come from the dataset of Lopez et al. [7]. In that study, the HOMO–LUMO values were obtained using DFT calculations.
>
> ### Question3：Application of the Lopez dataset
>
> We thank the reviewer for raising this question. Unlike prior work such as DeepAcceptor, we do not use the Lopez et al. (2017)[7] dataset directly for acceptor generation, nor do we treat its D–π–A structures as targets for molecular design. Instead, the dataset is used exclusively for pretraining the PCE predictor inside the Experimenter module.
>
> Specifically, in our PCE prediction model, the Lopez dataset serves as a large scale source of quantum chemical features that enable contrastive pretraining across molecular modalities. During this pretraining stage, the model learns aligned representations using HOMO/LUMO values from Lopez as auxiliary supervision signals. These DFT-computed energy levels help the model capture fundamental electronic trends, which in turn improves downstream prediction accuracy on OSC acceptor data.
>
> Crucially, the generative part of OSCAgent never uses Lopez data, and therefore no donor-oriented D–π–A structural patterns are injected into the generator. The pretrained Experimenter is later finetuned entirely on OSC acceptor datasets, which mitigates domain mismatch by adapting the learned representations to the specific distribution of real NFA acceptors.
>
> ### Question4：Handling conflicting evaluation signals
> When the Generator proposes a molecule that exhibits a severe conflict in the Experimenter’s evaluation , the Planner explicitly rejects such candidates. The Planner then guides the Generator to adjust subsequent designs in directions that mitigate the identified issue. Importantly, these conflicting molecules are not added to the Candidate Database, ensuring that they do not influence later retrieval or reinforce undesirable design patterns.
>
> [6]InternAgent: When Agent Becomes the Scientist – Building Closed-Loop System from Hypothesis to Verification, arxiv 2505.16938
>
> [7]Design principles and top non-fullerene acceptor candidates for organic photovoltaics, Joule 2017

---

### Official Review · Reviewer_Hqtz · 2025-11-01

**Soundness:** 3
**Presentation:** 3
**Contribution:** 2
**Rating:** 4
**Confidence:** 4

**Summary:**

The authors introduce a multi-agent framework for organic solar cell (OSC) molecule discovery. OSCAgent uses three collaborative agents: a planner, generator, and experimenter. The planner uses prior knowledge, reference databases, and candidate databases to create a generation plan. It also uses previous experiment reports to refine previous plans. The generator takes the generated plans to propose a new candidate. The experimenter uses cheminformatics tools and property prediction models to evaluate the proposed candidate and generate evaluation reports. The authors also present a new model to predict uncertainty-aware power conversion efficiency.

**Strengths:**

1. The proposed method is technically sound and provides an interesting pipeline to generate novel molecules
2. The retrieval-augmented strategy is novel and provides an interesting methodology to incorporate knowledge into generative models for novel OSC generation
3. Ablation studies show that the proposed methods with the RAG component and the experimenter improve the overall capabilities of the model

**Weaknesses:**

1. The baselines for both generative pipelines and predictive models and not using SOTA generative methods. BRICS and VAE models have suspiciously low validity.
2. Considering the high cost of running GPT-5 and other frontier reasoning models, the scalability of the proposed method may be a concern. What are the wall time/cost per predicted molecule?
3. The baseline high-performing OSC models are not detailed, so it is difficult to ascertain whether the proposed method substantially improved known molecules

**Questions:**

- How many molecules were generated using OSCAgent to evaluate the generation quality?

---

> ### Author Response · Authors · 2025-11-21
> **Response to Reviewer  Hqtz (1/2)**
>
> We thank the reviewer for the positive assessment of our work. We are glad that the reviewer finds our method technically sound and the overall pipeline for novel molecule generation interesting.
>
> Below, we provide detailed responses to the reviewer’s concerns.
> ### Weaknesses1 ：Additional baselines
> To address the reviewer’s concern regarding the choice of baselines, we have substantially expanded our experimental comparisons beyond those used in prior OSC molecular design studies.
>
> Most existing OSC design works rely almost exclusively on genetic algorithms (GAs), while diffusion-based generative models have not been systematically explored in this domain due to limited and highly specialized OSC datasets. To provide a more representative and competitive set of baselines, we included two widely used GA methods[4] (Smiles GA and Graph GA) and evaluated them directly against OSCAgent under identical scoring and evaluation conditions.
>
> To further strengthen the comparison, we incorporated a modern graph diffusion model, DiGress[1], as an additional SOTA generative baseline. DiGress was trained on the same molecular dataset used by DeepAcceptor[3] for its VAE baseline to ensure strict consistency and fairness across generative approaches.
>
> We also added an optimization-oriented latent-space method, the weighted retraining (WR) strategy proposed by Tripp et al.[2]. WR iteratively updates a generative model by reweighting samples based on objective values, shifting probability mass toward high-scoring regions of chemical space. For fidelity, we trained WR using the authors’ official implementation.
>
> Across all these enhanced comparisons, which span GA-based approaches, diffusion models, and latent-space optimization methods, OSCAgent consistently generates molecules with higher predicted PCE, better synthetic accessibility, and markedly improved chemical validity.
>
> | Model     | Uniqueness | Novelty | Validity | Avg. PCE (%) | Morgan  | MACCS  | RDK    | ECFP6  |
> |-----------|------------|---------|----------|--------------|---------|--------|--------|--------|
> | WR    | 80.83      | **100**     | 0.053    | 6.034        | 0.2144  | 0.5481 | 0.5686 | 0.1771 |
> | DiGress   | **90.13**      | **100**     | 0.001    | 4.541        | 0.2135  | 0.5627 | 0.5842 | 0.1433 |
> | Smiles GA | 62.57      | 86.65   | 0.112    | 7.275        | 0.2678  | 0.5965 | 0.8263 | 0.2389 |
> | Graph GA  | 88.47      | 82.27   | 0.180    | 7.543        | 0.3814  | 0.6064 | 0.6406 | 0.3262 |
> | **OSCAgent**  | 89.27      | **100**     | **0.705**    | **14.59**        | **0.4752**  | **0.7475** | **0.8576** | **0.3953** |
>
> It is also important to note that the limited size and diversity of available OSC datasets pose intrinsic challenges for diffusion models, significantly restricting their expressive capacity and their ability to generate chemically meaningful OSC-specific structures. In contrast, OSCAgent combines domain knowledge, retrieval-augmented structural priors, and the reasoning capabilities of large language models with tool usage, enabling effective OSC molecular design even under data-limited conditions and avoiding the low-validity issues observed in BRICS/VAE sampling.
>
> [1]DiGress: Discrete Denoising diffusion for graph generation, ICLR 2023.
>
> [2]Sample-Efficient Optimization in the Latent Space of Deep Generative Models via Weighted Retraining, NIPS 2020
>
> [3]Accelerating the discovery of acceptor materials for organic solar cells by deep learning, npj computational materials.
>
> [4]Sample Efficiency Matters: A Benchmark for Practical Molecular Optimization, NIPS 2022.

---

> ### Author Response · Authors · 2025-11-21
> **Response to Reviewer Hqtz (2/2)**
>
> ### Weaknesses2 and Question1 : Scalability and computational cost of OSCAgent
>
> We thank the reviewer for raising these important questions regarding scalability and evaluation. In our experiments using the GPT-5 API, OSCAgent requires on average 14,595 input tokens and 1,109 output tokens to generate a single molecule, corresponding to an approximate cost of 0.029 USD per molecule, with a typical end-to-end wall-clock time of 25 seconds. For the main experiments reported in the paper, we generated approximately 1,500 OSC molecules, which were then used to evaluate validity, synthetic accessibility, and PCE-related metrics.
>
> We also emphasize that the framework is not dependent on GPT-5. As demonstrated in Appendix E.1, OSCAgent operates effectively with open-source GPT-OSS models, indicating that the workflow, which combines retrieval guided planning and chemistry aware generation, is model agnostic and can be scaled to more cost efficient language models without compromising generation quality.
>
> ### Weaknesses3: Clarification of the set of high-performing OSC molecules
> We appreciate the reviewer’s comment and understand the need for clearer baseline definitions. In our study, we follow the criteria commonly adopted in prior OSC literature and DeepAcceptor, defining high-performing OSC acceptors as those with experimentally reported PCE > 10% and reasonable synthetic accessibility (SAscore < 8). Using these criteria, we curated a benchmark set of 464 high-performing OSC molecules, which serves as the reference against which OSCAgent-generated molecules are evaluated.
>
> In the revised manuscript, we will include explicit details of this benchmark set.
>
> Using this benchmark as the reference, we find that OSCAgent consistently produces molecules of higher overall quality than other baseline generative methods. More than 70% of the generated molecules satisfy key OSC design criteria, and their structural similarity distribution (computed using standard molecular fingerprints such as ECFP4/Morgan) is substantially closer to that of the benchmark set compared to GA-, VAE-, or diffusion-based baselines. This indicates that OSCAgent not only generates chemically plausible structures but also explores chemical space in a manner more aligned with known high-performing OSC acceptors.

---

### Official Review · Reviewer_DoQv · 2025-11-01

**Soundness:** 4
**Presentation:** 3
**Contribution:** 3
**Rating:** 6
**Confidence:** 3

**Summary:**

This paper proposed a framework to use LLM Agent to consolidate and automate the process of planning, generating and validating new organic solar cell molecules. The proposed framework uses RAG to retrieve related information to form a plan, and use that to generate candidate molecules. After that, another agent will supervise the process of validation of these molecules. The result will then return to the planner agent to summarize, preparing for the next loop. This paper also proposed a model to predict the power conversion efficiency of a given molecule graph, SMILES string and molecular fingerprint.

**Strengths:**

The PCE prediction model is innovative and peroformative in this setting, and combines the graph and SMILES string and the molecular fingerprint provides a new way to think about evaluation because traditionally we only focus on either side when it comes to predicting molecule properties.

The experiment demonstrates superior performance compared to other models, indicating that LLM agents are capable of using retrieved knowledge and applying it to the generation process.

The paper is well-written, covering all necessary details. Furthermore, the ablation study is sound and serves as strong proof.

**Weaknesses:**

1. Agentic research framework is basically the trend on pretty much every area, and limiting it to only generate organic solar cells seems to be only to testament that the framework is working, not to propose anything entirely new.
2. There seems to be no discussion on the computation resources and time used for the agent compared to traditional methods. Ideally, they would save time if the model would generate all valid molecules in one shot with the expertise knowledge guided. But the more likely scenario is that many failed generated molecules are discarded during the reflection process, which ultimately give you better results, but at the expense of actual multiple runs of generation. I believe using a script to run a traditional generation method and select with the given predictor would also achieve a relatively good result, without an LLM involvement.
3. In the experiment, the model used for comparison seems to be quite outdated. Recent research advancement indicates that diffusion models are pretty much capable to generate de-nuovo molecules that satisfy design criteria, and have significantly better performance compared to old models.

**Questions:**

What’s the limitation for this model?
Did you try any other molecule generation models in comparison with your model?

---

> ### Author Response · Authors · 2025-11-21
> **Response to Reviewer DoQv (1/3)**
>
> We sincerely thank the reviewer for the thoughtful evaluation of our submission and for the encouraging remarks regarding the soundness and clarity of our work. We thank you for your constructive comments.
>
> Below, we provide detailed responses to the reviewer’s concerns.
>
> ### Weaknesses1 ：Novelty of the agentic framework
> We appreciate the reviewer’s concern regarding the novelty of applying an agentic framework to a specific domain. We respectfully clarify that OSCAgent is not a direct application of a generic agent paradigm, but introduces domain-specific architectural, algorithmic, and chemical innovations that are essential for OSC molecular discovery and do not arise from agentic structure alone. each component in OSCAgent is designed specifically for OSC molecular discovery, incorporating domain knowledge, constraints and physics-informed signals unavailable in standard agent frameworks:
>
> **1、A retrieval-augmented design strategy specific to OSCs, not a generic RAG module**
> Our retrieval design is chemistry-aware and cannot be swapped with off-the-shelf RAG:
>
> The Reference Database is curated from experimentally validated high-performing OSC acceptors and pruned with structural diversity guarantees using K-center greedy selection (Algorithm 1)
>
> The Candidate Database is dynamically updated with a chemistry-grounded scoring function that jointly rewards predicted PCE, penalizes SAscore, and enforces orbital-energy feasibility windows (Appendix C.2)
>
> This retrieval strategy encodes domain knowledge that typical agent frameworks lack completely.
>
> **2、Domain-specific multi-agent roles tightly coupled to chemical workflows**
> Unlike general-purpose agentic systems, the Planner–Generator–Experimenter triad in OSCAgent is explicitly designed around OSC-specific chemistry tasks, each with domain-motivated responsibilities:
>
>
> Planner performs retrieval-augmented analysis of experimentally validated high-performance acceptors and dynamically updated top candidates, using chemical fingerprints and K-center greedy structural coverage (Appendix C.1)
>
> Experimenter conducts chemical validity checks, HOMO/LUMO modeling, SAscore estimation, and uncertainty-aware PCE prediction (Section 4.2; Appendix D)
>
> The Generator performs chemically constrained structural design guided by the Planner’s retrieved knowledge and strategic instructions, rather than engaging in unconstrained free-form text generation.
>
> This cooperative workflow is tailored to OSC molecular physics and constraints, not a generic agent pipeline.
>
> **3、A PCE Prediction Model Specifically Designed for OSC Molecules**
> In addition, OSCAgent introduces a specialized PCE prediction model tailored specifically for OSC acceptor molecules. The model integrates **(i)** graph-based molecular representations, **(ii)** SMILES-based encoders, and **(iii)** handcrafted molecular fingerprints through a MOE architecture, and is trained with a chemistry-motivated, multimodal pretraining strategy  before fine-tuning on experimental OSC data. Furthermore, the model explicitly incorporates uncertainty-aware regression, treating PCE as a Gaussian-distributed random variable—a crucial consideration due to the inherent experimental variability of OSC measurements.
>
> This domain-specific design significantly outperforms both traditional ML baselines and recent GNN/Graph Transformer variants on the OSC dataset. The predictor is therefore not a generic surrogate, but a dedicated OSC-oriented modeling contribution.
>
> These components produce functionally new capabilities that generic agent frameworks do not offer.

---

> ### Author Response · Authors · 2025-11-21
> **Response to Reviewer DoQv (2/3)**
>
> ### Weaknesses2 ：Computational efficiency and comparison to traditional generative pipelines
> We agree with the reviewer that, on a per-molecule basis, an LLM-based agentic pipeline is inherently slower than lightweight rule-based recombination (e.g., BRICS) or VAE latent-space sampling. Traditional methods can indeed decode millions of structures within seconds, whereas an LLM performs sequential reasoning and text generation.
>
> However, raw OSC molecular generation speed is not the dominant bottleneck in OSC molecular discovery. In conventional workflows, the primary time cost arises not from generating molecules but from the extensive downstream filtering required to compensate for the extremely low signal-to-noise ratio of these generative processes. For example, DeepAcceptor generated over 4.8 million structures via BRICS/VAE sampling, yet the vast majority were chemically invalid, synthetically infeasible, or far outside the OSC design distribution—necessitating substantial post-processing and aggressive filtering. Thus, while per-molecule decoding is fast, the effective cost per useful molecule is extremely high.
>
> In contrast, OSCAgent explicitly prioritizes quality over volume. During generation, the planner integrates:
> • retrieval-augmented OSC specific structural motifs,
> • HOMO/LUMO feasibility windows grounded in physical constraints,
> • synthetic accessibility control via SAscore, and
> • chemically constrained structural design rather than unconstrained SMILES sampling.
> Consequently, OSCAgent maintains over 70% chemically valid and synthetically feasible outputs, significantly improving the effective yield relative to brute-force pipelines.
>
> OSCAgent mirrors the way human chemists design molecules: knowledge-guided reasoning → focused, purposeful exploration → a small set of high-quality candidates. In practice, what scientists need is a few genuinely promising structures, rather than a large number of randomly generated molecules with limited relevance or utility.
>
> Finally, despite LLM reasoning steps, the end-to-end cost remains practical for research workflows. In our experiments, OSCAgent generates a high-quality candidate in ~25 seconds, which is still orders of magnitude faster than traditional expert-driven OSC design cycles that typically span days.

---

> ### Author Response · Authors · 2025-11-21
> **Response to Reviewer DoQv (3/3)**
>
> ### Weaknesses3 and Questions1：Additional baselines
> We have substantially expanded our experimental comparisons beyond those used in prior OSC molecular design studies.
>
> Most existing OSC design works rely almost exclusively on genetic algorithms (GAs), while diffusion-based generative models have not been systematically explored in this domain, primarily due to limited data availability. To ensure a fair and representative comparison, we included two widely used GA methods[4](Smiles GA ,Graph GA) as additional baselines and evaluated them directly against OSCAgent.
>
>
> To further address the reviewer’s concern, we incorporated a modern graph diffusion model, DiGress[1], as an additional baseline. We trained DiGress on the same molecular dataset used by DeepAcceptor[3] for its VAE baseline to ensure consistent and fair comparison conditions across all generative approaches.
>
> In addition, we also incorporated optimization-oriented latent-space methods, such as the weighted retraining (WR)[2] strategy proposed by Tripp et al. This method iteratively updates a generative model by reweighting samples according to their objective values, thereby gradually shifting probability mass toward higher-scoring regions of the chemical space. We trained the model using the authors’ official implementation.
>
> | Model     | Uniqueness | Novelty | Validity | Avg. PCE (%) | Morgan  | MACCS  | RDK    | ECFP6  |
> |-----------|------------|---------|----------|--------------|---------|--------|--------|--------|
> | WR    | 80.83      | **100**     | 0.053    | 6.034        | 0.2144  | 0.5481 | 0.5686 | 0.1771 |
> | DiGress   | **90.13**      | **100**     | 0.001    | 4.541        | 0.2135  | 0.5627 | 0.5842 | 0.1433 |
> | Smiles GA | 62.57      | 86.65   | 0.112    | 7.275        | 0.2678  | 0.5965 | 0.8263 | 0.2389 |
> | Graph GA  | 88.47      | 82.27   | 0.180    | 7.543        | 0.3814  | 0.6064 | 0.6406 | 0.3262 |
> | **OSCAgent**  | 89.27      | **100**     | **0.705**    | **14.59**        | **0.4752**  | **0.7475** | **0.8576** | **0.3953** |
>
> Across all these comparisons, OSCAgent consistently produced molecules with higher predicted PCE, better synthetic accessibility, and greater chemical validity than GA-based and diffusion-based baselines.
>
> Due to the limited size and diversity of available OSC datasets, training diffusion models remains challenging, which restricts their expressive capacity and limits their ability to generate chemically meaningful structures. In contrast, OSCAgent combines domain knowledge with the reasoning capabilities of large language models and tool usage, enabling effective OSC molecular design even under limited data conditions.
>
> The main limitation of our current framework is that the generated molecules are evaluated entirely in silico and therefore represent potentially promising candidates rather than experimentally validated OSC materials.  In future work, we plan to integrate more specialized materials simulations and collaborate with wet-lab experts to incorporate experimental feedback, enabling iterative refinement and validation of the designed molecules.
>
>
> [1]DiGress: Discrete Denoising diffusion for graph generation, ICLR 2023.
>
> [2]Sample-Efficient Optimization in the Latent Space of Deep Generative Models via Weighted Retraining, NIPS 2020
>
> [3]Accelerating the discovery of acceptor materials for organic solar cells by deep learning, npj computational materials.
>
> [4]Sample Efficiency Matters: A Benchmark for Practical Molecular Optimization, NIPS 2022.

---

### Author Response · Authors · 2025-12-02
**Summary of Revision**

Dear Chairs and Reviewers：
We would like to thank the reviewers for their careful and constructive comments. We also thank the reviewers for recognizing the originality and effectiveness of our framework (DoQv, Hqtz), the novelty of our  design strategy (Hqtz), and the strong potential of our agentic pipeline for advancing autonomous OSC discovery (UQiv, F5D9). The paper has been revised in accordance with the reviewers’ comments and suggestions. Updates and changes are marked by blue color in the revised version.

The major changes in this revision lie in the following aspects:

Appendix C: Data Processing and Modeling Strategy

In this section, we clarify our modeling objective and provide a explanation of the data processing and modeling strategy, demonstrating the soundness and rationale behind the choices made in our framework.

Appendix D.4: K Value Selection

In this section, we examine the impact of hyperparameters used in the retrieval stage.

Appendix F.3: Additional Baselines

Appendix F.4: Computational Cost of OSCAgent

Appendix F.5: Case Studies

Here we provide additional generated OSC molecules, as shown in Fig.9 and Fig.10.


Should you need further information, please let us know. We look forward to hearing from you soon.

Yours sincerely,
Authors of Paper “OSCAgent: Closing the Loop in Organic Solar Cell Discovery with LLM Agents”

---

### Note · Authors · 2026-01-26

**Comment:**

I have read and agree with the venue's withdrawal policy on behalf of myself and my co-authors.

**Withdrawal Confirmation:**

I have read and agree with the venue's withdrawal policy on behalf of myself and my co-authors.

---

### Meta-Review · Area_Chair_Bucr · 2025-12-20

**Summary:**

This paper presents an LLM-based multi-agent framework for automated organic solar cell molecule design. The direction is timely and the high-level concept is interesting, and several reviewers appreciated the authors’ detailed rebuttal and additional experiments.

However, despite these strengths, the reviews highlight persistent concerns regarding the novelty of the proposed framework, the scientific soundness and realism of the problem setup, and the lack of concrete evidence for computational efficiency and scalability. Questions also remain about baseline completeness and whether the approach provides clear advantages beyond simpler or more modern generative pipelines. While the rebuttal improved clarity, the core issues were not fully resolved, and reviewer opinions remained mixed.

**Reviewer Concerns:**

Reviewer DoQv raised concerns regarding the novelty and scope of the proposed agentic framework, arguing that agent-based workflows are now widespread and that restricting the application to organic solar cells does not demonstrate a substantive methodological contribution. They also questioned the computational efficiency of the approach and the absence of quantitative comparisons to simple traditional generation–selection pipelines, noting that the rebuttal’s additions still lacked concrete runtime metrics and did not fully address whether non-LLM pipelines could perform similarly. In terms of baseline coverage, the reviewer pointed out that the evaluation relied on outdated generative models and should include more modern diffusion-based generators; although the authors expanded baselines and provided explanations, the justification for excluding diffusion models remained unconvincing. Overall, despite thoughtful clarifications in the rebuttal, the reviewer’s central concerns regarding novelty, practical utility, and efficiency remained largely unresolved.

---

Reviewer Hqtz raised concerns regarding the baseline selection, noting that the generative and predictive baselines (e.g., BRICS and VAE) were not state-of-the-art and showed unusually low validity. The reviewer also questioned the scalability and computational cost of using GPT-5–level models, asking for concrete wall-time or cost per generated molecule. Additional issues included insufficient detail on the high-performing OSC models used as references and a lack of clarity on how many molecules were generated to assess OSCAgent’s design quality. In the rebuttal, the authors added several baselines, clarified the computational cost and scalability considerations, and provided details on the OSC reference models. These additions helped address several of the reviewer’s concerns, though the scalability issue may still require more extensive evidence.

---

Reviewer UQiv raised fundamental concerns about the scientific validity of the problem formulation, emphasizing that predicting PCE solely from the acceptor molecule, without specifying or assuming a donor counterpart, renders the task conceptually unsound given that OSC efficiency critically depends on donor–acceptor interactions. They also noted that several generated molecules appeared chemically unrealistic, lacking symmetry typically required for viable non-fullerene acceptors, and pointed out technical issues such as PDF display errors. Additional comments addressed overstatements regarding “closed-loop discovery,” unclear positioning relative to existing LLM-based scientific agent systems, and missing methodological details, including donor assumptions, HOMO–LUMO data sources, use of the Lopez dataset, and handling of conflicting evaluation signals. In the rebuttal, the authors clarified the rationale for focusing on acceptor molecules, defended the generation of asymmetric acceptor architectures, corrected PDF and typographical issues, and provided clarifications regarding closed-loop claims, dataset usage, energy-level sourcing, and evaluator consistency. These responses improved clarity, the reviewer’s core concern regarding the scientific soundness of the task itself remains substantial.

----

Reviewer F5D9 engaged in an in-depth discussion with the authors. After the first rebuttal, some issues were resolved, but several substantive concerns remained. The reviewer noted that parts of the framework, particularly the scoring function and design choices, still appeared somewhat arbitrary or overly complex, raising the possibility of overfitting to the specific setup. They appreciated the addition of new iterative baselines and found the comparisons encouraging, but pointed out inconsistencies in the interpretation of synthetic accessibility (SAscore), questioning whether higher or lower values were actually desirable and whether claims about OSCAgent’s advantages were aligned with the rebuttal. Furthermore, they emphasized that if fingerprints play a key role, stronger baselines using fingerprint-augmented learned representations should be included. Following the authors’ further explanations, particularly regarding the scoring function design, the interpretation of SAscore changes, and fingerprint-based baselines.

**Reviewer Scores:**

Reviewer DoQv is a nice reviewer. After the rebuttal, it is likely that they will maintain their original score, although a downward adjustment cannot be fully ruled out.

---

Reviewer Hqtz may maintain its original score.

---

Reviewer UQiv may moderately improve their score after the rebuttal.

---

Reviewer F5D9  is likely to maintain their original score, though an upward adjustment is possible given the constructive dialogue and partial resolution of their concerns

---

### Decision · Program_Chairs · 2026-01-26

Reject